# CRISPR screens reveal genetic determinants of PARP inhibitor sensitivity and resistance in prostate cancer

Takuya Tsujino[1,2,11], Tomoaki Takai[1,2,11], Kunihiko Hinohara[3,4,11], Fu Gui [1], Takeshi Tsutsumi [1,2], Xiao Bai[1], Chenkui Miao[1], Chao Feng[1], Bin Gui[1], Zsofia Sztupinszki[5,6], Antoine Simoneau[7], Ning Xie[8], Ladan Fazli[8], Xuesen Dong [8,9], Haruhito Azuma[2], Atish D. Choudhury [3], Kent W. Mouw[10], Zoltan Szallasi [5,6], Lee Zou[7], Adam S. Kibel[1] & Li Jia [1] ✉

Prostate cancer harboring *BRCA1/2* mutations are often exceptionally sensitive to PARP inhibitors. However, genomic alterations in other DNA damage response genes have not been consistently predictive of clinical response to PARP inhibition. Here, we perform genome-wide CRISPR-Cas9 knockout screens in BRCA1/2-proficient prostate cancer cells and identify previously unknown genes whose loss has a profound impact on PARP inhibitor response. Specifically, *MMS22L* deletion, frequently observed (up to 14%) in prostate cancer, renders cells hypersensitive to PARP inhibitors by disrupting RAD51 loading required for homologous recombination repair, although this response is *TP53*-dependent. Unexpectedly, loss of *CHEK2* confers resistance rather than sensitivity to PARP inhibition through increased expression of BRCA2, a target of CHEK2-TP53-E2F7-mediated transcriptional repression. Combined PARP and ATR inhibition overcomes PARP inhibitor resistance caused by *CHEK2* loss. Our findings may inform the use of PARP inhibitors beyond BRCA1/2-deficient tumors and support reevaluation of current bio-markers for PARP inhibition in prostate cancer.

Despite treatment advances, metastatic castration-resistant prostate cancer (mCRPC) remains a lethal disease. Genomic sequencing studies reveal that approximately 90% of mCRPC patients carry clinically actionable genomic alterations[1]. Alterations in genes involved in the DNA damage response (DDR) are among the most common genetic events. Approximately 10% of primary and 25% of metastatic prostate cancer (PCa) patients have an alteration in at least one gene involved in DDR[2], which represents a potential therapeutic vulnerability. In particular, defects in homologous recombination repair (HRR) render cells highly sensitive to inhibition of Poly (ADP-ribose) polymerase (PARP). As a targeted therapy, PARP inhibitors (PARPis) prevent PARP1 and PARP2 from repairing DNA single-strand breaks (SSBs) and lead to stalled and collapsed replication forks by trapping PARP1 and PARP2 on the DNA breaks[3]. Subsequently, SSBs are converted to

[1]Division of Urology, Department of Surgery, Brigham and Women's Hospital & Harvard Medical School, Boston, MA, USA. [2]Department of Urology, Osaka Medical and Pharmaceutical University, Osaka, Japan. [3]Department of Medical Oncology, Dana-Farber Cancer Institute & Harvard Medical School, Boston, MA, USA. [4]Department of Immunology, Nagoya University Graduate School of Medicine, Nagoya, Japan. [5]Computational Health Informatics Program, Boston Children's Hospital, Boston, MA, USA. [6]Danish Cancer Society Research Center, Copenhagen, Denmark. [7]Department of Pathology, Massachusetts General Hospital & Harvard Medical School, Boston, MA, USA. [8]Vancouver Prostate Centre, Vancouver General Hospital, Vancouver, British Columbia, Canada. [9]Department of Urologic Sciences, University of British Columbia, Vancouver, British Columbia, Canada. [10]Department of Radiation Oncology, Dana-Farber Cancer Institute & Brigham and Women's Hospital & Harvard Medical School, Boston, MA, USA. [11]These authors contributed equally: Takuya Tsujino, Tomoaki Takai, Kunihiko Hinohara. ✉e-mail: ljia@bwh.harvard.edu

double-strand breaks (DSBs) that HRR-deficient cells cannot repair effectively, leading to overwhelming DNA damage, cell cycle arrest, and cell death. The *BRCA1/2* genes encode proteins essential for HRR, a pathway that repair DNA DSBs. PARPis selectively induce synthetic lethality in cancer cells harboring mutations in the *BRCA1/2* genes[4,5].

Several PARPis are presently under clinical investigation in PCa, including olaparib, rucaparib, niraparib, and talazoparib, as a single agent. Two of them (olaparib and rucaparib) have been approved by the U.S. Food and Drug Administration (FDA) for the treatment of mCRPC patients with deleterious germline and/or somatic mutations in *BRCA1/2*, and the olaparib indication includes mutations in 12 additional HRR genes[6–11]. It becomes clear that BRCA1/2-dficient tumors exhibit high sensitivity and improved outcome to PARP inhibition based on results from these trials, as measured by standard radiographic criteria, 50% decrease in prostate-specific antigen (PSA), circulating tumor-cell counts, and progression-free survival or overall survival. However, whether and to what extent PARPis can be used to treat tumors with non-*BRCA1/2* alterations remains controversial after gene-by-gene analysis. Furthermore, HRR-deficient tumors are not always sensitive to PARPis. Both intrinsic and acquired resistance to PARP inhibition represents a formidable clinical problem. Therefore, one of the major barriers to effective treatment using PARPis is distinguishing patients who may and may not benefit from PARP inhibition. The genome-wide CRISPR-Cas9 (clustered regularly interspersed short palindromic repeat-CRISPR associated nuclease 9) knockout (KO) screen is a powerful and unbiased approach to identify genes that, when deleted, confer PARPi sensitivity or resistance. Previous studies using CRISPR screens in non-PCa cell lines have identified genes whose loss impacts PARPi response[12–14].

In this work, we carry out CRISPR screens in four BRCA1/2-proficient PCa cell lines with the goal of expanding the use of PARPis beyond *BRCA1/2* mutation carriers and finding undefined synthetically lethal interactions. Our screens lead to the identification of *MMS22L*, a frequently deleted HRR gene in PCa, as a predictive biomarker for PARPis. Surprisingly, we find that loss of *CHEK2* causes PARPi resistance, which we explore using a therapeutic approach through ATR inhibition to overcome in pre-clinical models.

## Results

### Genome-wide CRISPR KO screens identify genes that modulate PARPi response in PCa cells

To identify genes whose loss increases or decreases the sensitivity of PCa cells to PARP inhibition, we performed genome-wide CRISPR KO screens in the PCa LNCaP, C4-2B, 22Rv1 and DU145 cells in the presence of olaparib or DMSO vehicle. These four cell lines reflect different aspects of PCa progression to castration resistance and have no predicted biallelic deleterious mutations in *BRCA1/2* and other canonical HRR genes determined by whole exome sequencing (Supplementary Data 1). LNCaP and C4-2B cells are relatively more sensitive to olaparib in contrast to 22Rv1 and DU145 cells (Supplementary Fig. 1), but less sensitive when compared to *BRCA1*-null ovarian cancer UWB1.289 cells. Cells were transduced with the lentiviral-based CRISPR-Cas9 KO libraries, targeting over 18,000 protein-coding genes as previously described[15,16], followed by 28 days of treatment with olaparib or DMSO (Fig. 1a; Supplementary Fig. 2a, b). The abundance of single guide RNA (sgRNA) for each gene was assessed by β-score and the differential β-score was further calculated using the MAGeCKFlute pipeline by comparing olaparib to DMSO treatment[17,18]. We identified 216, 243, 153, and 211 negatively selected genes (with a stringent cutoff of *p*-value <0.01), the loss of which sensitizes cells to olaparib in LNCaP, C4-2B, 22Rv1, and DU145 cells, respectively (Fig. 1b; Supplementary Data 2). There were 67 genes shared by at least two cell lines, considered as common hits (Supplementary Data 3). Gene Ontology (GO) analyses revealed that

these common hits were enriched for DNA repair and replication functions (Fig. 1c). Further analyses of each individual cell line showed that the mitochondrial complex I assembly was the most enriched function in DU145 (Fig. 1d; Supplementary Fig. 3). Dysfunction of mitochondrial complex I causes a decline of NAD$^+$ and ATP supply required for PARP-mediated DNA repair[19]. This cell-type specific mechanism may provide a unique therapeutic vulnerability.

We identified 280, 347, 222, and 281 positively selected genes (*p* < 0.01), the loss of which renders cells resistant to olaparib in LNCaP, C4-2B, 22Rv1, and DU145 cells, respectively (Fig. 1e; Supplementary Data 4). These genes are more likely related to intrinsic resistance rather than acquired resistance arising after prolonged PARP inhibition. There were 103 genes shared in at least two cell lines (Supplementary Data 5). These common hits were enriched in cell cycle phase transition and positive regulation of gene expression (Fig. 1f). Analyses of positively selected genes in each individual cell line showed that cell cycle genes are largely enriched in CRPC C4-2B, 22Rv1, and DU145 cells, but not in androgen-dependent LNCaP cells (Fig. 1d; Supplementary Fig. 3), reflecting the role of cell cycle transition in mediating PARPi resistance in CRPC[20].

To further dissect the identified genes implicated in PARPi sensitivity and resistance, we analyzed the common hits using the database in the STRING protein-protein association network[21]. We found that 50% of the common negatively selected genes could be assigned to well-defined functions in DDR, including HRR, DNA replication, Fanconi anemia, helicase, and cohesion (Fig. 1g). In contrast, approximately 50% of the common positively selected genes were involved specifically in cell cycle, ADP-ribosylation, and transcriptional regulation (Fig. 1h). Finally, we ranked all genes based on the average of differential β-scores from all four cell lines. Representative top-ranked negatively and positively selected genes are shown in Fig. 1i, including ones studied in this work. Notably, the *BRCA1/2* genes were highly ranked in negative selection but reached the cutoff only in C4-2B and DU145 cells, respectively. This is likely because acute inactivation of BRCA1/2 is lethal without adaptive mechanisms in cells[22–24], as reflected by the depletion of sgRNAs targeting *BRCA1/2* genes even in the absence of olaparib treatment, thereby failing to reach statistically significant depletion under the olaparib treatment condition (Supplementary Fig. 4). The same trend was observed for the *PALB2* gene. Together, our screen results provide a global view of genetic determinants of PARPi response in PCa.

### Validation of negatively selected genes related to PARPi sensitivity

Next, we examined the genomic alteration (deletion and mutation) frequency of the 67 common negatively selected genes in the TCGA (the primary tumor) and SU2C/PCF (metastatic tumor) cohorts (Fig. 2a)[25,26]. We selected ten genes, based on their alteration frequency as well as functions in DDR, and individually deleted these genes in C4-2B cells (Supplementary Fig. 5a). We observed markedly increased sensitivity to olaparib following genetic deletion (Fig. 2b). Among these genes, *HELLS* and *WDR76* are the only two hits in all four cell lines although they are not among the most frequently deleted or mutated genes in PCa. We further deleted *HELLS* or *WDR76* in 22Rv1 and DU145 cells, leading to significantly increased response to olaparib (Supplementary Fig. 5b). HELLS is a helicase and chromatin remodeling enzyme that promotes the initiation of HRR and contributes to DSB repair[27], while WDR76 is a specific reader of 5-hydroxymethylcytosine (5hmC) with undefined functions in DNA repair[28,29]. Indeed, WDR76 has a large interaction network, including HELLS and PARP1[30], supporting synthetically lethal interaction with PARP inhibition. *RNASEH2B* and *MMS22L* are located on chromosomes 13q14 and 6q16, respectively, which have long been recognized as two frequently deleted regions in PCa[31–37]. While loss of *RNASEH2B* might cause ribonucleotide excision

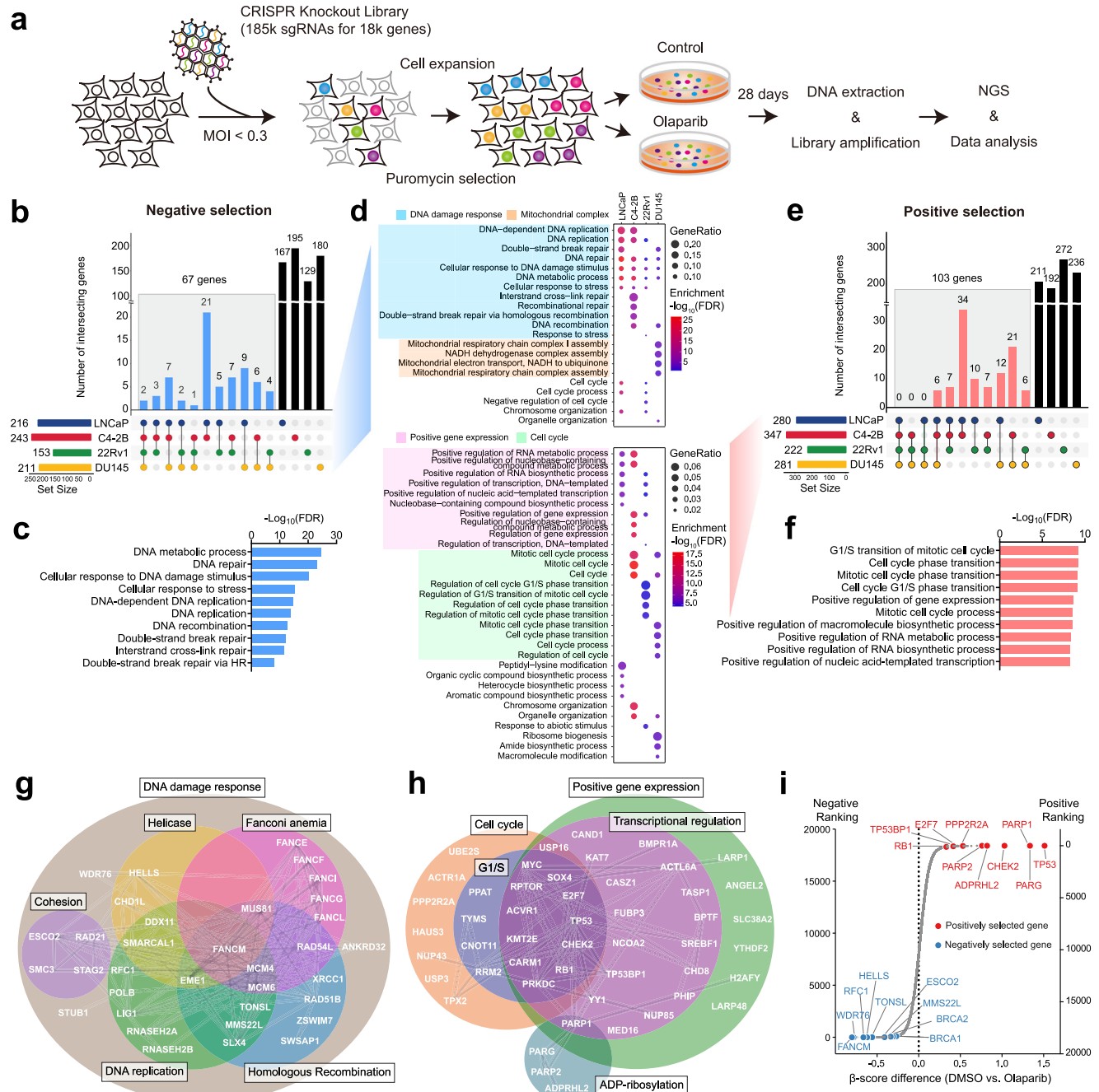

**Fig. 1 | CRISPR screens identify genes that modulate PARPi response in PCa cells. a** Schematic of genome-wide CRISPR/Cas9 screens. **b** UpSet plot[110] of negatively selected genes in four PCa cell lines as indicated. Blue bars indicate the number of common hits in at least two screens. **c** Top GO terms enriched in 67 common hits from negative selection. **d** Top GO terms enriched in negatively (upper panel) and positively (lower panel) selected genes in each individual cell line. **e** UpSet plot of positively selected genes in four PCa cell lines as indicated. Red bars indicate the number of common hits in at least two screens. **f** Top GO terms enriched in 103 common hits from positive selection. **g** The networks of common hits from negative selection grouped according to their roles in specific pathways and their genetic and physical interactions (gray lines) based on STRING analysis. **h** The networks of common hits from positive selection, grouped as in (**g**). **i** Top-ranked genes from CRISPR screens determined by comparing olaparib to DMSO treatment. Genes are ranked by the average of differential β-scores from all four cell lines. Negatively and positively selected genes are marked in blue and red, respectively.

repair deficiency and PARP-trapping lesions as previously reported[13], we recently reported that co-loss of *RB1*, a closely located tumor suppressor gene, was antagonistic in PCa cells[38]. MMS22L is known to form an obligate heterodimer complex with TONSL, which reads histone H4 unmethylated lysine 20[39] and promotes HRR of stalled or collapsed replication forks and attendant DSBs by aiding RAD51 loading[40–42]. Cells with *MMS22L* deletion are highly sensitive to the topoisomerase I inhibitor camptothecin. However, the impact of *MMS22L* loss on PARP inhibition has not been studied.

## Loss of *MMS22L* increases PARPi response due to impaired HRR function in PCa cells

Since MMS22L always forms a complex with TONSL for replication-associated DNA damage repair, TONSL was unsurprisingly identified as one of the top hits as well (Fig. 1i; Supplementary Data 2). Nevertheless, the high frequency of *MMS22L* homozygous deletion (14% in primary and 5% in metastatic prostate tumors) makes it an attractive biomarker to predict PARPi response in PCa. To further validate this finding, we deleted *MMS22L* and *TONSL* in five PCa cell lines - LNCaP, C4-2B, 22Rv1,

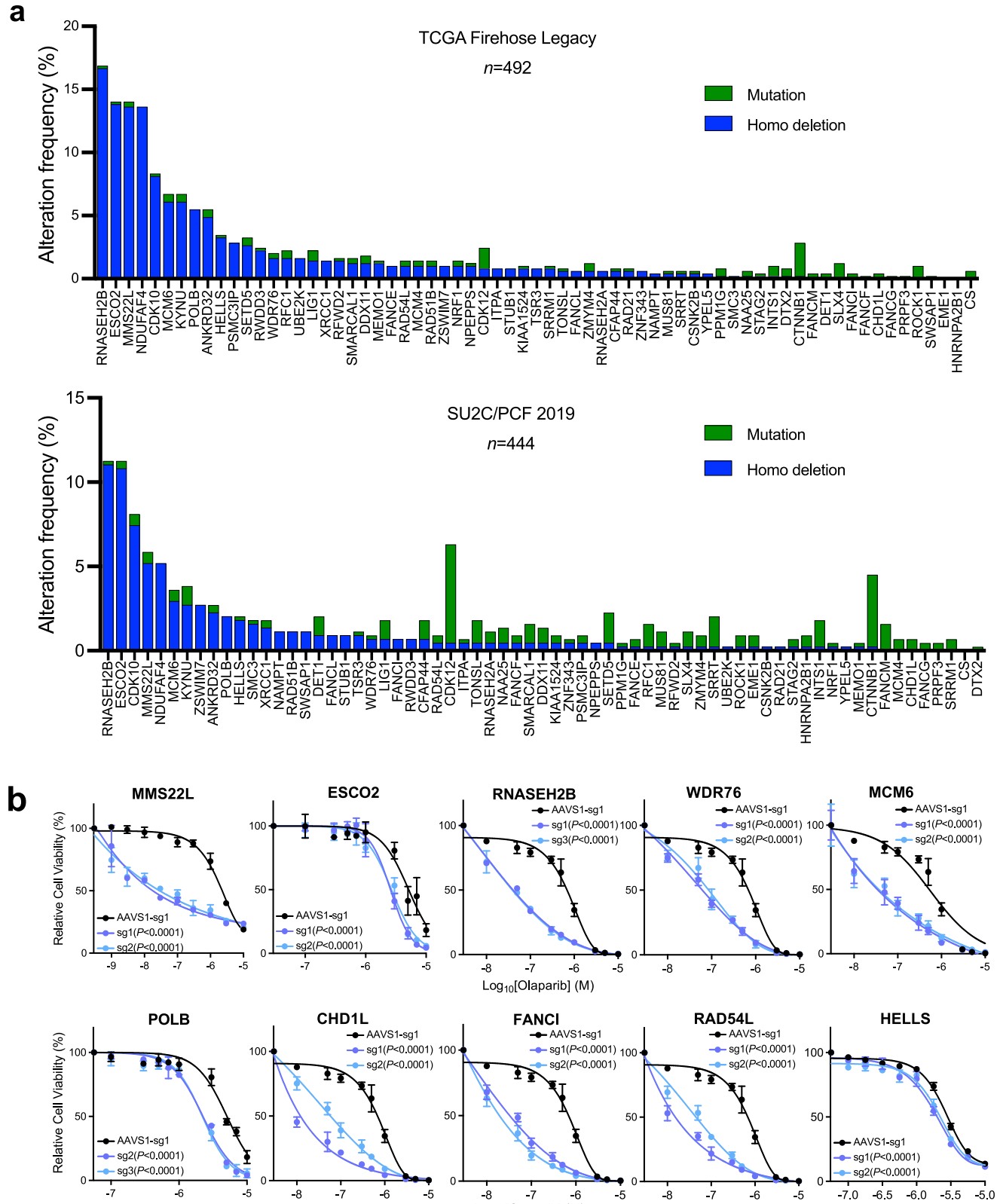

**Fig. 2 | Validation of negatively selected genes with frequent genomic alterations. a** The frequency of mutations and homozygous (Homo) deletions in 67 common negatively selected genes from the TCGA Firehose Legacy cohort (Upper panel, $n$ = 492) and the SU2C/PCF cohort (Lower panel, $n$ = 444)[25,26]. **b** Dose-response curves after treatment with olaparib for C4-2B cells with gene knockout (KO) as indicated. Data are presented as mean ± SD ($n$ = 3 biologically independent experiments). The $p$-values were determined by comparing two gene-specific sgRNAs to a control AAVS1 sgRNA using two-way ANOVA. Source data are provided as a Source Data file.

PC-3, DU145, and MDAPCa2b (Fig. 3a). We showed that deletion of either *MMS22L* or *TONSL* resulted in significantly increased sensitivity to olaparib in LNCaP, C4-2B, MDAPCa2b and 22Rv1 cells, but not in DU145 or PC-3 cells. In a growth competition experiment, we observed that olaparib treatment significantly reduced the fraction of *MMS22L*-KO cells in contrast to corresponding control cells when they were equally pre-mixed and grown under the same treatment condition (Fig. 3b). Notably, *MMS22L*-deleted C4-2B cells were also sensitive to other PARPis (rucaparib, talazoparib, and veliparib), but displayed lower sensitivity to carboplatin, a DNA crosslinking agent (Fig. 3c). In addition, we generated *MMS22L*-KO C4-2B single cell clones through cell sorting and confirmed either complete (sg1, clone #1-4) or partial *MMS22L* deletion (sg1, clone #5-6; sg2, clone #1-6) using immunoblot (Fig. 3d). All *MMS22L*-KO clones exhibited high sensitivity to olaparib to a similar extent without a gene dose effect. Restoration of MMS22L expression completely abolished the sensitivity of *MMS22L*-KO cells to olaparib (Fig. 3e), confirming that the phenotype was specifically due to the loss of *MMS22L*.

Next, we sought to investigate the mechanism by which loss of *MMS22L* increased PARPi sensitivity. A recent study showed that PARPi induces replication fork collapse and DSBs in a trans cell cycle manner and BRCA1/2-deficient cells cannot recruit RAD51 to repair them, leading to cell death[20]. Similarly, we observed increased expression of γ-H2AX and cleaved-PARP in *MMS22L*-KO C4-2B cells following 72-hour olaparib treatment (Fig. 4a), reflecting PARPi-induced DNA DSBs and apoptosis. Increased γ-H2AX foci were detected by immunofluorescence staining (Fig. 4b). Cell cycle analysis showed a substantial accumulation of cells in G2/M phase after deletion of *MMS22L* or *TONSL* (Fig. 4c and Supplementary Fig. 6). Given the function of the MMS22L-TONSL complex in loading RAD51 at DSB sites and promoting HRR after replication fork collapse independent of BRCA2[40], we postulated that MMS22L loss might impair HRR that occurred predominantly in the late S and G2 phases of the cell cycle, leading to overwhelming DSBs and mitotic catastrophe. Indeed, using immunofluorescence staining, we found significantly reduced RAD51 foci in *MMS22L*-KO C4-2B cells after olaparib treatment for 24 h in comparison with control cells (Fig. 4d). RAD51 loading was enhanced after MMS22L expression was restored. From these results, we conclude that MMS22L loss confers high sensitivity to PARPis likely due to compromised RAD51 recruitment to PARPi-induced DSBs, causing homologous recombination deficiency (HRD).

Tumors with HRD have been found to display specific patterns of genomic alterations (or genomic scars), which can be quantified based on the genome-wide measurement of three patterns of genomic instability (loss of heterozygosity, telomeric allelic imbalance, and large-scale state transitions)[43]. A high HRD score (≥42) has been shown to be predictive of clinical benefits with PARPi or platinum therapy in breast and ovarian cancer patients[44,45]. Using whole-genome sequencing data, we previously showed that a subset of PCa patients with a high HRD score did not harbor germline or somatic mutations in *BRCA1/2* and other canonical HRR genes[46]. Further analysis revealed that tumors with heterozygous or homozygous *MMS22L* deletion displayed significantly higher HRD scores (Fig. 4e), although homozygous deletion had much higher scores. And the highest HRD score was observed in tumors with both BRCA and MMS22L deficiency. The *MMS22L* loss-mediated genomic instability was also confirmed using HRDetect (Supplementary Fig. 7), a model that quantitatively aggregates six HRD-associated mutation signatures[47]. These results may partially explain previously unclarified cause of HRD in PCa. In addition, using publicly available PCa clinical data, we found *MMS22L* deletion is correlated with decreased *MMS22L* transcript levels (Fig. 4f). Tumors with heterozygous or homozygous deletion had significantly lower *MMS22L* mRNA levels compared to those with wild-type *MMS22L*. Patients with lower *MMS22L* expression had a shorter survival (Fig. 4g),

suggesting tumor progression driven by HRD-mediated genomic instability.

## The response to PARP inhibition after *MMS22L* loss is *TP53*-dependent

Next, we asked why deletion of *MMS22L* did not increase PARPi response in PC-3 and DU145 cells (Fig. 3a). CRISPR screens revealed *TP53* as one of the top resistance genes in LNCaP, C4-2B, and 22Rv1 cells (Fig. 1i; Supplementary Data 4). This was validated by genetic deletion of *TP53* in these cell lines (Fig. 5a). LNCaP and C4-2B cells possess wild-type *TP53*, while 22Rv1 cells harbor a monoallelic *TP53* mutation, leading to a relatively low level of p53 protein expression (Supplementary Data 1 and Supplementary Fig. 8). In contrast, PC-3 cells have *TP53* truncating mutations and do not express p53 (p53-null), and DU145 cells harbor dominant negative *TP53* mutations[48]. Notably, PCa cell lines with *TP53* mutations are generally less sensitive to olaparib compared to cell lines with wild-type *TP53* (Supplementary Fig. 1). Furthermore, RNA-sequencing (RNA-seq) analysis revealed 271 commonly upregulated genes in LNCaP and C4-2B cells after olaparib treatment, in which the p53 pathway is significantly enriched (Fig. 5b, c; Supplementary Data 6), supporting the role of p53 activation in modulating the response to PARP inhibition. We reasoned that *TP53* status might impact PARPi response of *MMS22L*-deleted cells. Using an RNA interference approach, we showed that knockdown of *MMS22L* expression significantly increased PARPi sensitivity of *TP53* wild-type C4-2B cells, whereas *TP53*-KO cells remained insensitive (Fig. 5d). Conversely, knockdown of *TP53* expression significantly reduced PARPi sensitivity of *MMS22L*-KO C4-2B cells with less DSBs (γ-H2AX foci) detected (Supplementary Fig. 9a, b). Using a Dox-inducible system, we reintroduced the *TP53* gene into p53-null PC-3 cells. Restoration of p53 expression re-sensitized *MMS22L*-KO PC-3 cells to olaparib in vitro and in vivo (Fig. 5e, f). In contrast, restoration of AR expression in *AR*-negative PC-3 cells had no effect on PARPi response after *MMS22L* knockdown (Supplementary Fig. 10). Furthermore, we observed that the level of γ-H2AX protein expression and foci were increased after restoration of p53 expression in *MMS22L*-KO PC-3 cells, indicating more DNA DSBs (Fig. 5g, h). Accordingly, cell apoptosis was increased as determined by cleaved PARP levels. These results suggest that *MMS22L* deletion increases PARPi sensitivity only when p53 remains functional. Approximately 12.6% of primary and 3.8% of metastatic prostate tumors harbor *MMS22L* homozygous deletion with wild-type *TP53* (Fig. 5i). The percentage will increase to 27% and 20.5% respectively when heterozygous deletion is included. To further verify *MMS22L* deletion in clinical samples, we utilized DNAscope assay, a chromogenic DNA in situ hybridization technique, on a tissue micro-array (TMA) with 146 primary PCa tissue cores. We detected 4.1% homozygous deletion and 26% heterozygous deletion of *MMS22L* in primary tumors (Fig. 5j; Supplementary Fig. 11). Together, our results suggest that loss of *MMS22L* occurs in a considerable fraction of PCa patients, who may therefore benefit from PARP inhibition.

## Validation of positively selected genes related to PARPi resistance

Next, we set out to further investigate PARPi resistance. We found that the top positively selected genes include those previously identified as conferring PARPi resistance in other contexts, including *PARP1*, *PARP2*, *PARG*, *ADPRHL2* (also known as *ARH3*), and *TP53BP1* (Fig. 1i)[3,49–52]. Although not genetically altered as frequently as *TP53* in PCa, down-regulation of these genes could influence PARPi response. *PPP2R2A* was previously included in the HRR gene list used to select patients for the PROfound trial (NCT02987543). However, the FDA subsequently removed *PPP2R2A* from the gene panel because outcomes appeared worse for patients with *PPP2R2A* alterations treated with olaparib compared to the control[6]. In accordance with the clinical result, we identified *PPP2R2A* as one of the common hits from positive selection

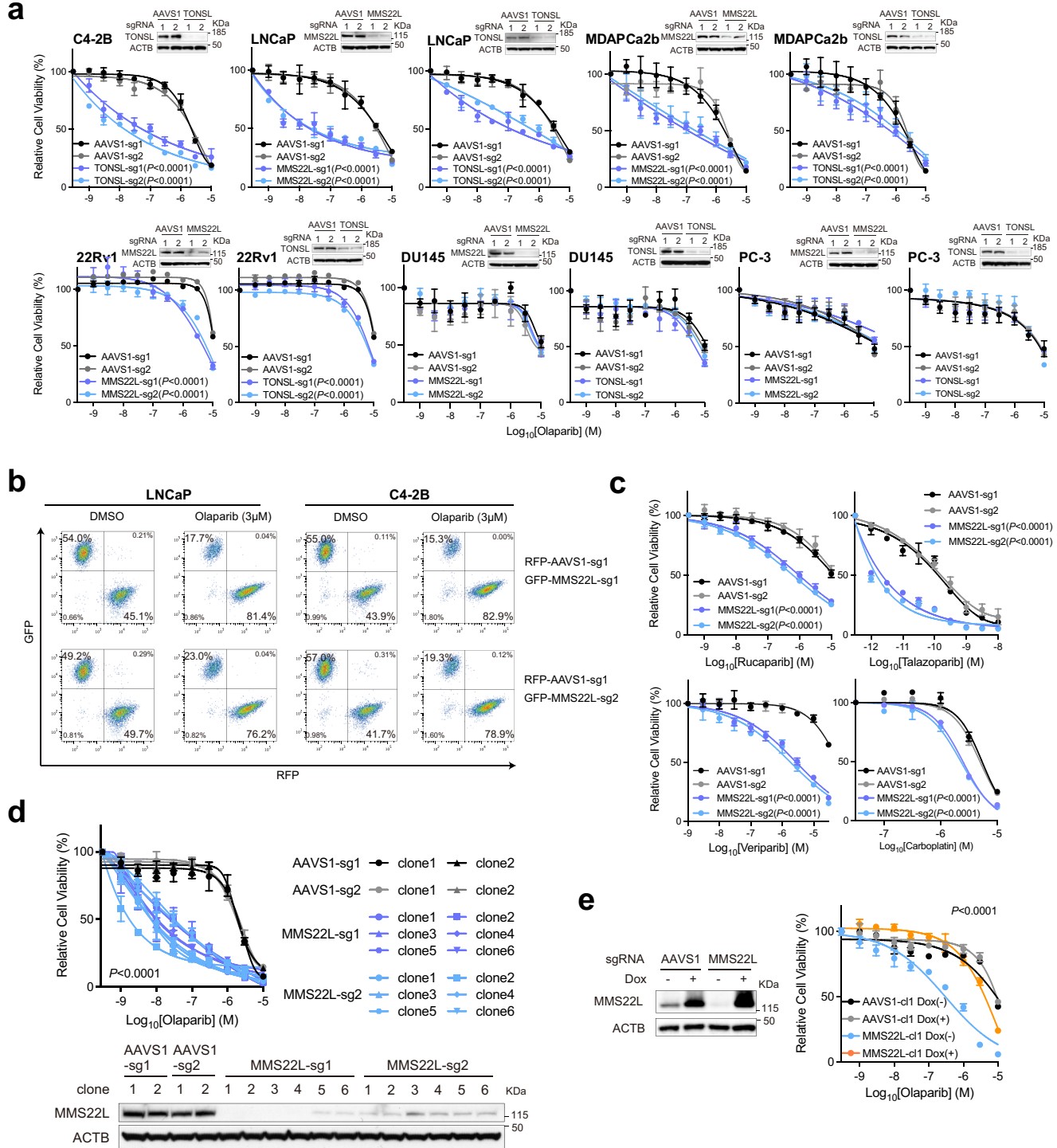

**Fig. 3 | Loss of *MMS22L* increases PARPi response in PCa cells. a** Dose-response curves after treatment with olaparib for *MMS22L*- or *TONSL*-KO cells versus corresponding AAVS1 control cells of C4-2B, LNCaP, MDAPCa2b, 22Rv1, DU145 and PC-3. Upper right panels are immunoblot analysis of *MMS22L* or *TONSL* KO efficiency. ACTB (β-actin) is a loading control. The olaparib response of *MMS22L*-KO C4-2B cells is presented in (**2b**). **b** Flow cytometry analysis of GFP and RFP positive cells. *MMS22L*-KO LNCaP or C4-2B cells (with GFP) were co-cultured with corresponding AAVS1 control cells (with RFP) in a 1:1 ratio in the presence of DMSO or olaparib. Two *MMS22L*-KO cell lines (sg1 and sg2) and one control cell line (sg1) were used. Cells were collected and analyzed using flow cytometry 7 days after the treatment. The percentage of each cell population is presented in each panel. The experiment was repeated independently twice with similar results. **c** Dose-response curves after treatment with rucaparib, talazoparib, veliparib, and carboplatin for two *MMS22L*-

KO C4-2B cell lines (sg1 and sg2) versus two AAVS1 control cell lines (sg1 and sg2). **d** Dose-response curves (upper panel) after treatment with olaparib for AAVS1 control and *MMS22L*-KO C4-2B cell clones. Immunoblot analysis (lower panel) showing the MMS22L protein level in AAVS1 control and *MMS22L*-KO cell clones. **e** Immunoblot analysis (left panel) showing the MMS22L protein level in C4-2B AAVS1 control sg1 clone 1 (cl1) and *MMS22L*-KO sg1 clone 1 (cl1), stably infected with TET-inducible sgRNA-resistant *MMS22L* gene, after treatment with or without doxycycline (0.15 μg/ml) for 3 days. Dose-response curves (right panel) after olaparib treatment with or without doxycycline (0.15 μg/ml) treatment for the same C4-2B cell clones. In **a** and **c**–**e** data are presented as mean ± SD (*n* = 3 biologically independent experiments). The immunoblot analyses were repeated independently twice with similar results. The *p*-values were determined by two-way ANOVA. Source data are provided as a Source Data file.

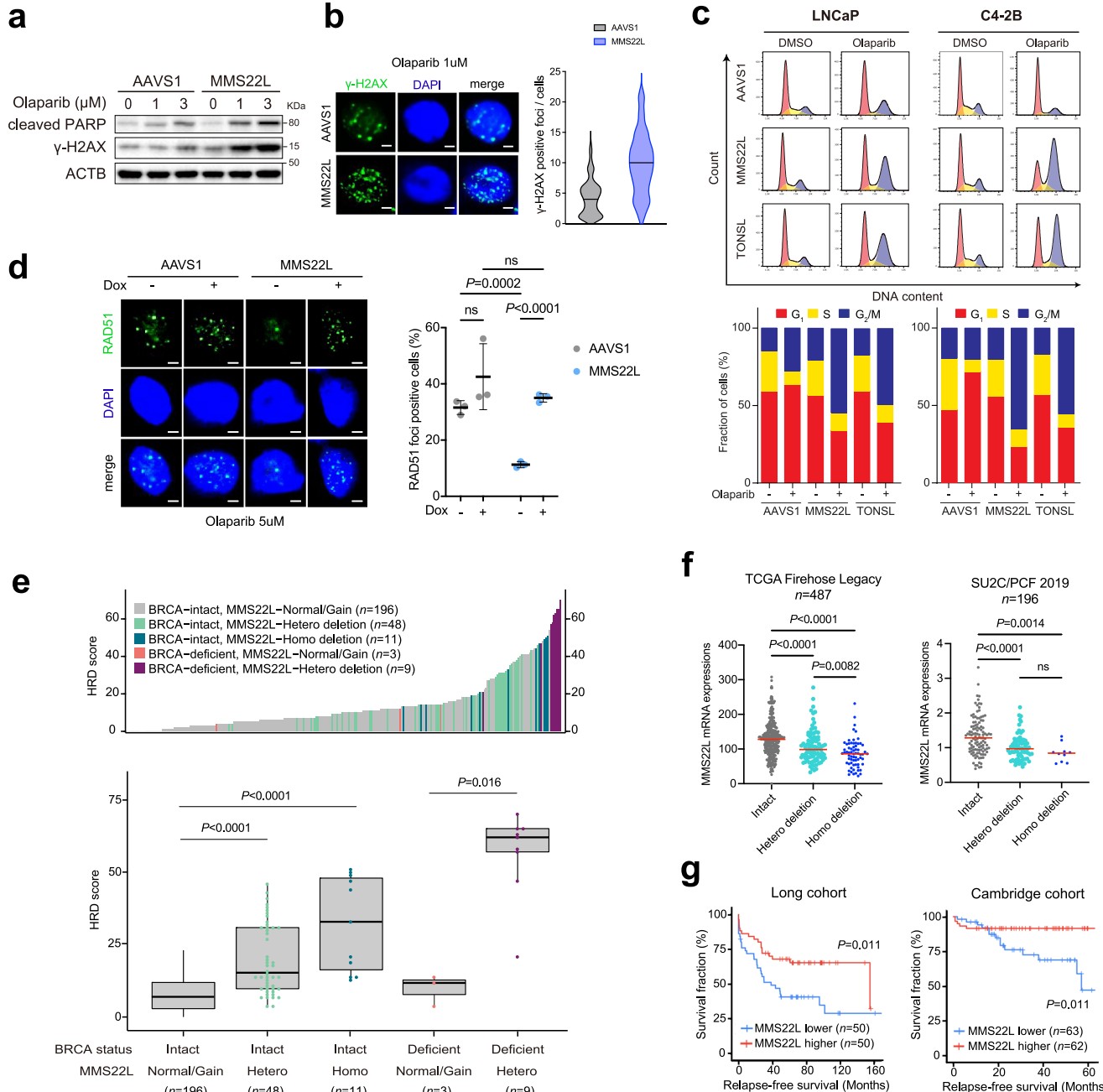

**Fig. 4 | Loss of *MMS22L* impairs HRR function in PCa cells. a** Immunoblot analysis of cleaved PARP and γ-H2AX in AAVS1 control and *MMS22L*-KO C4-2B cells after olaparib treatment for 72 h. The experiment was repeated independently three times with similar results. **b** Representative images of two biologically independent experiments and quantification of γ-H2AX foci in AAVS1 control and *MMS22L*-KO C4-2B cells after olaparib treatment for 24 h. More than 100 cells were analyzed per condition. Solid lines inside the violin indicate the median. Scale bar = 5 μm. **c** Cell cycle analysis (upper panel) of AAVS1 control, *MMS22L*-KO and *TONSL*-KO LNCaP, and C4-2B cells after treatment with DMSO or olaparib for 72 h. The percentage of cells (lower panel) in each phase of the cell cycle is shown. The experiment was repeated independently twice with similar results. **d** Representative images and quantification of RAD51 foci in AAVS1 control and *MMS22L*-KO C4-2B cells stably infected with TET-inducible *MMS22L* gene after olaparib treatment in the presence or absence of doxycycline (0.15 μg/ml) for 24 h. Dots indicate each replicate with more than 100 cells analyzed. Data are presented as mean ± SD of three biologically

independent replicates. Scale bar = 5 μm. **e** Ranked HRD scores (Upper panel) in PCa tumors with *BRCA* and/or *MMS22L* genomic alterations as indicated. Comparison of HRD scores (lower panel) between five patient groups classified based on *BRCA* and *MMS22L* status (*n* = 196, 48, 11, 3, and 9 tumor samples in each group, respectively). Data are presented as boxplot indicating median, 25th-75th percentile (box), and minimum and maximum values (whiskers). **f** The mRNA level of *MMS22L* in PCa tumors with Intact *MMS22L*, heterozygous (Hetero) deletion, and homozygous (Homo) deletion of *MMS22L* in the TCGA cohort and the SU2C/PCF cohort (*n* = 487 and 196 tumor samples, respectively)[25,26]. **g** Kaplan–Meier survival curves in the Long PCa cohort[111] (*n* = 50 versus 50 tumor samples) and the Cambridge PCa cohort[112] (*n* = 63 versus 62 tumor samples) based on the *MMS22L* mRNA expression level (lower versus higher). A log-rank test was carried out to examine the survival difference. In (**b**) and (**d**–**f**) the *p*-values were determined using two-sided *t*-test. ns = not significant. Source data are provided as a Source Data file.

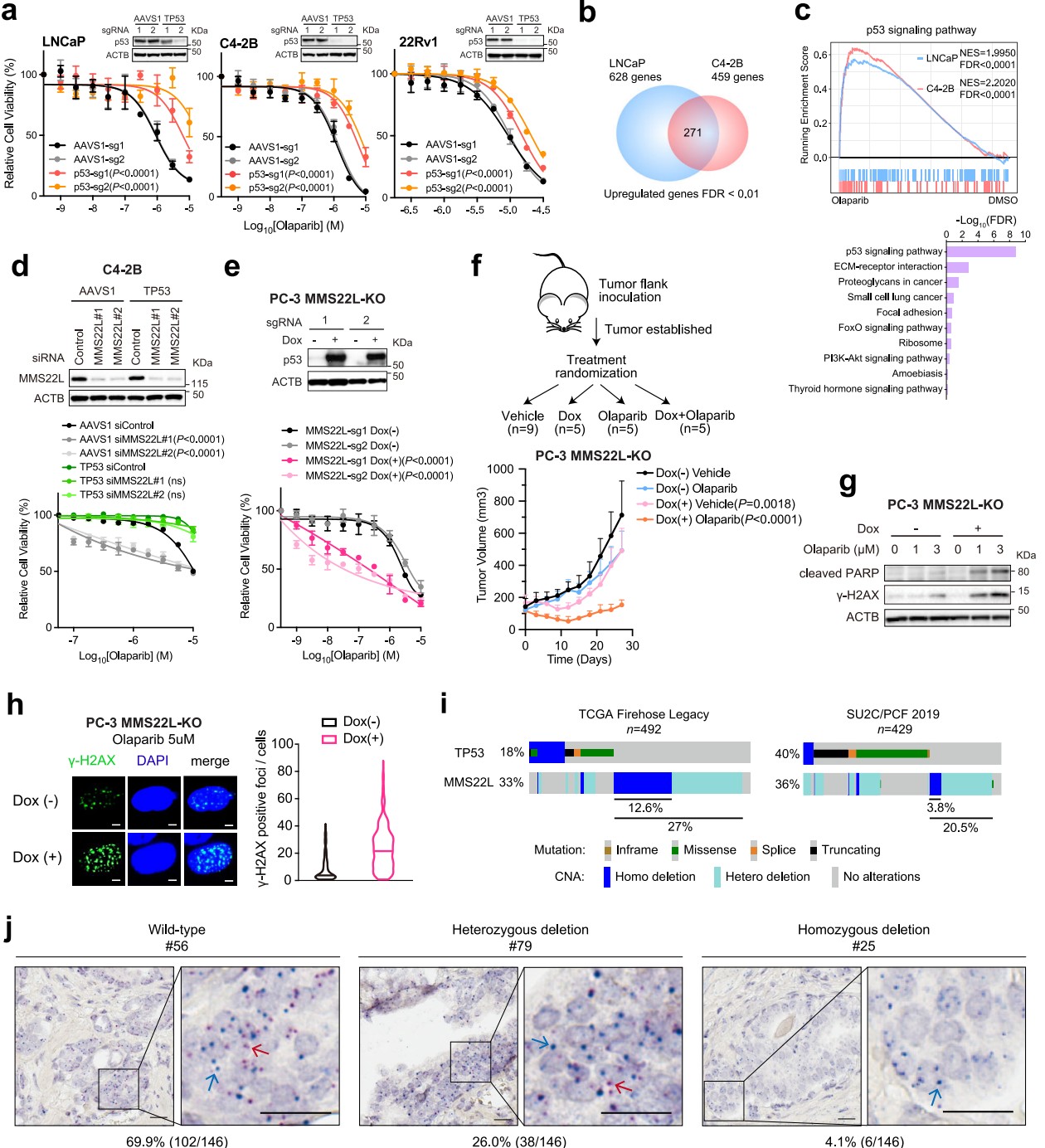

**Fig. 5 | *TP53* status impacts PARPi response in *MMS22L*-depleted PCa cells.**
**a** Dose-response curves after treatment with olaparib for the indicated cells after *TP53* deletion. Immunoblot analyses of p53 in *TP53*-KO versus control cells are shown. **b** Venn diagram of up-regulated genes in LNCaP and C4-2B cells after olaparib treatment. **c** GSEA (upper panel) and top enriched KEGG pathways (lower panel) of up-regulated genes. Normalized enrichment score (NES) and false discover rate (FDR) are indicated. **d** Immunoblot analysis (upper panel) of MMS22L in AAVS1 control and *TP53*-KO C4-2B cells after siRNA knockdown. Dose-response curves (lower panel) after olaparib treatment for the same cells with or without *MMS22L* siRNA transfection. **e** Immunoblot analysis (upper panel) of p53 in AAVS1 control and *MMS22L*-KO PC-3 cells containing TET-inducible *TP53* gene in the presence or absence of doxycycline (0.15 μg/ml) for 3 days. Dose-response curves (lower panel) after olaparib treatment in the presence or absence of doxycycline. **f** Schematic diagram of the experimental procedure (upper panel). Tumor growth (lower panel) after treatment with olaparib (50 mg/kg) or vehicle with or without doxycycline induction (*n* = 9, 5, 5, and 5 mice in each group, respectively). Data are presented as mean ± SD. The *p*-values were determined by comparing between

treatment with and without doxycycline using two-way ANOVA. **g** Immunoblot analysis of cleaved PARP and γ-H2AX in cells after the treatment as described in (**e**). The experiment was repeated independently three times with similar results. **h** Representative images of two biologically independent experiments and quantification of γ-H2AX foci after olaparib treatment for 24 h in the presence or absence of doxycycline (0.15 μg/ml). More than 100 cells were analyzed per condition. Solid lines inside the violin indicate the median. Scale bar = 5 μm. **i** The frequency of *TP53* and *MMS22L* alterations in the TCGA and the SU2C/PCF cohorts[25,26]. **j** Representative images of *MMS22L* wild-type, heterozygous and homozygous deletion determined by DNAscope assay using a tissue microarray (n = 146 tissue cores). Red signals (red arrow) indicate probes targeting the *MMS22L* gene on chromosome 6q. Blue signals (blue arrow) indicate control probes targeting the centromeric region. Scale bar = 20 μm. In (**a**) and (**d–e**), data are presented as mean ± SD (*n* = 3 biologically independent experiments). The immunoblot analyses were repeated independently twice with similar results. The *p*-values were determined using two-way ANOVA. Source data are provided as a Source Data file.

(Fig. 1i; Supplementary Data 4). *PPP2R2A* is located on chromosome 8p and frequently deleted in PCa (Supplementary Fig. 12). Indeed, chromosome 8p is the most frequently lost chromosomal arm of all tumors (36%) in the TCGA cohort[37,53,54], suggesting a potentially common intrinsic resistance mechanism in tumors with 8p loss.

Surprisingly, we identified Checkpoint kinase 2 (*CHEK2*) as one of the top resistance genes in LNCaP, C4-2B, and 22Rv1 cells. This is an unexpected result since *CHEK2* is considered as a BRCAness gene[55] and an FDA-approved biomarker for olaparib treatment in mCRPC patients. *CHEK2* has been used as a biomarker of PARPi sensitivity in several clinical trials[7–10,56,57] because of its known function in promoting HRR through phosphorylation of BRCA1[58,59]. We deleted *CHEK2* in five PCa cell lines and observed significantly reduced olaparib response in LNCaP, C4-2B, and 22Rv1 cells, but not in *TP53*-mutant PC-3 and DU145 cells using cell viability and colony formation assays (Fig. 6a, b). We further generated *CHEK2*-KO C4-2B single-cell clones, which all displayed resistance to olaparib (Fig. 6c). The resistance was also observed with other PARPis (rucaparib, talazoparib, and veliparib) and carboplatin in C4-2B and 22Rv1 cells (Fig. 6d). We conclude that *CHEK2* loss is associated with PARPi resistance rather than sensitivity in PCa cells with functional p53.

## Loss of *CHEK2*/*TP53* enhances HRR function through E2F7-controlled BRCA2 expression

CHK2 kinase regulates multiple proteins beyond BRCA1 in response to DNA damage. CHK2 phosphorylates p53 on serine 20 and stabilizes p53, leading to cell cycle arrest in G1 phase[60,61]. We, therefore, asked whether PARPi resistance caused by *CHEK2* loss is mediated through p53 inactivation. In line with the activation of p53 pathway after PARP inhibition (Fig. 5c), we found that olaparib treatment increased phosphorylated CHK2 and total p53 protein expression levels in *CHEK2*-intact C4-2B and 22Rv1 cells, but not in *CHEK2*-KO cells (Fig. 7a). The extent of DNA DSBs was also greater in *CHEK2*-intact cells compared to *CHEK2*-KO cells following olaparib treatment as measured by γ-H2AX protein expression and foci (Fig. 7a, b). Accordingly, there was increased olaparib-induced apoptosis in *CHEK2*-intact cells as measured by cleaved-PARP levels (Fig. 7a). Deletion of *TP53* had no effect on olaparib-induced CHK2 phosphorylation but reduced DNA DSBs (γ-H2AX expression) and cell apoptosis (cleaved-PARP) as similarly observed in *CHEK2*-KO cells (Fig. 7c).

Next, we sought to investigate the CHEK2-TP53 downstream target genes that may contribute to PARPi resistance in *CHEK2*-KO cells. Previous studies have shown that *E2F7* is a *TP53*-reulated gene[62], and that deletion of *E2F7* renders cells resistant to PARP inhibition through upregulation of RAD51 in BRCA2-deficient cells[63]. *E2F7* was also one of the top positively selected genes in our CRISPR screens (Fig. 1i). E2F7 is an atypical member of the E2F transcription factor family, which regulates its target genes, including HRR genes, through transcription repression rather than activation. We reasoned that *CHEK2* loss might derepress HRR gene expression via the TP53-E2F7 axis. To investigate p53-mediated regulation of *E2F7* expression, we analyzed publicly available p53 ChIP-seq data in multiple cell lines and found highly enriched p53 binding immediately upstream of the *E2F7* transcription start site (TSS) (Supplementary Fig. 13a), indicating a conserved and direct transcriptional regulation mechanism. Using p53 ChIP-qPCR, we observed baseline p53 binding at the *E2F7* promoter, which was increased after olaparib treatment in *CHEK2*-intact C4-2B and 22Rv1 cells (Fig. 7d). However, olaparib-induced p53 binding was not observed after *CHEK2* loss. Accordingly, *E2F7* expression was increased after olaparib treatment in *CHEK2*-intact C4-2B cells within a few hours, but not in *CHEK2*-KO cells (Fig. 7e). Furthermore, the MDM2 inhibitor (MDM2i) nutlin, an agent that blocks MDM2-mediated ubiquitination and promotes p53 stabilization, increased p53 expression and p53 binding at the *E2F7* promoter as well (Supplementary Fig. 13b, c), supporting p53-regulated *E2F7* expression. Indeed, MDM2 inhibition

sensitized *CHEK2*-KO C4-2B cells to olaparib in line with the role of the TP53-E2F7 axis in PARPi resistance (Supplementary Fig. 13d).

To investigate E2F7-controlled HRR gene expression, we analyzed publicly available E2F7 ChIP-seq data in multiple cell lines and observed strong E2F7 binding immediately upstream of the TSSs of *BRCA1/2* and *RAD51* genes (Supplementary Fig. 13e). Using E2F7 ChIP-qPCR, we demonstrated strong E2F7 binding at these genomic regions in both *CHEK2*-intact and *CHEK2*-KO C4-2B and 22Rv1 cells (Fig. 7f). Importantly, E2F7 binding was increased in *CHEK2*-intact control cells after olaparib treatment but remained unchanged or decreased in *CHEK2*-KO cells. In agreement with the E2F7 ChIP results, we found that mRNA levels of *BRCA1/2* and *RAD51* were decreased after olaparib treatment in *CHEK2*-intact C4-2B cells but increased in *CHEK2*-KO cells (Fig. 7g), indicating E2F7-mediated transcriptional suppression on HRR genes. It should be noted that no significant changes of cell cycle and growth were observed after *CHEK2* or *TP53* deletion in C4-2B cells (Supplementary Fig. 14a, b). Therefore, the upregulation of *BRCA1/2* and *RAD51* gene expression is largely due to transcriptional regulation rather than cell cycle alteration, although the expression of HRR genes is cell cycle-dependent. The protein levels of BRCA1/2 and RAD51 were decreased as well after olaparib treatment in *CHEK2*-inatct cells (Fig. 7h). However, the protein level changes were not all consistent with the mRNA level changes in *CHEK2*-KO cells. While the BRCA2 and RAD51 protein levels were increased or remained at a high level following olaparib treatment, the BRCA1 protein level was decreased. This is likely because BRCA1 is directly regulated by CHK2 through posttranslational modification[58]. On the other hand, the reduction of BRCA1 protein expression was not observed in *TP53*-KO cells since p53 is a downstream effector of CHK2 (Supplementary Fig. 15). These results suggest that the CHEK2-TP53-E2F7 axis is activated with olaparib treatment, leading to suppression of HRR gene expression and that this axis is disrupted when either *CHEK2* or *TP53* is lost. Specifically, the BRCA2 protein level was significantly upregulated after *CHEK2* or *TP53* loss. In addition, we found that olaparib-induced BRCA2 suppression was rescued after knockdown of *E2F7*, rendering cell resistant to olaparib (Fig. 7i; Supplementary Fig. 16).

To determine whether HRR capacity was enhanced after *CHEK2* loss, we performed RAD51 foci formation assay and found significantly increased olaparib-induced RAD51 foci in *CHEK2*-KO C4-2B and 22Rv1 cells (Fig. 7j). These results suggest that the PARPi resistance arising from *CHEK2* loss is, at least in part, dependent on enhanced HRR function. Increased HRR capacity (RAD51 foci) and the resulting reduced DNA DSBs (γ-H2AX foci) were observed in the same *CHEK2*-KO cells (Supplementary Fig. 17a). To further determine whether increased BRCA2 expression is responsible for PARPi resistance after *CHEK2* loss, we knocked down BRCA2 in *CHEK2*-KO C4-2B and 22Rv1 cells and found that these cells were sensitized to olaparib (Fig. 7k). Similarly, knockdown of BRCA2 in *TP53*-KO cells increased olaparib sensitivity (Supplementary Fig. 17b). In agreement with these results, we found a negative correlation between CHEK2 and BRCA2 protein expression in the TCGA cohort (Fig. 7l)[25]. Together, our results suggest that the CHEK2-TP53-E2F7-BRCA2 pathway is likely one of the primary mechanisms for PARP inhibitor resistance after *CHEK2* loss (Fig. 7m).

## ATR inhibition overcomes PARPi resistance in *CHEK2*-deficient PCa cells

Finally, we sought to explore therapeutic strategies to overcome PARPi resistance after *CHEK2* loss. Emerging evidence has shown that PARPi resistance is often accompanied by increased ATR activity, which coordinates cell cycle checkpoint response through phosphorylation of CHK1 and allows cells to survive PARPi-induced replication stress[64]. We found that ATR activity was elevated in *CHEK2*-KO cells with olaparib treatment, as evidenced by increased CHK1 phosphorylation in a dose-dependent manner (Supplementary Fig. 18). We treated *CHEK2*-KO cells with olaparib in combination with M6620, a clinically used

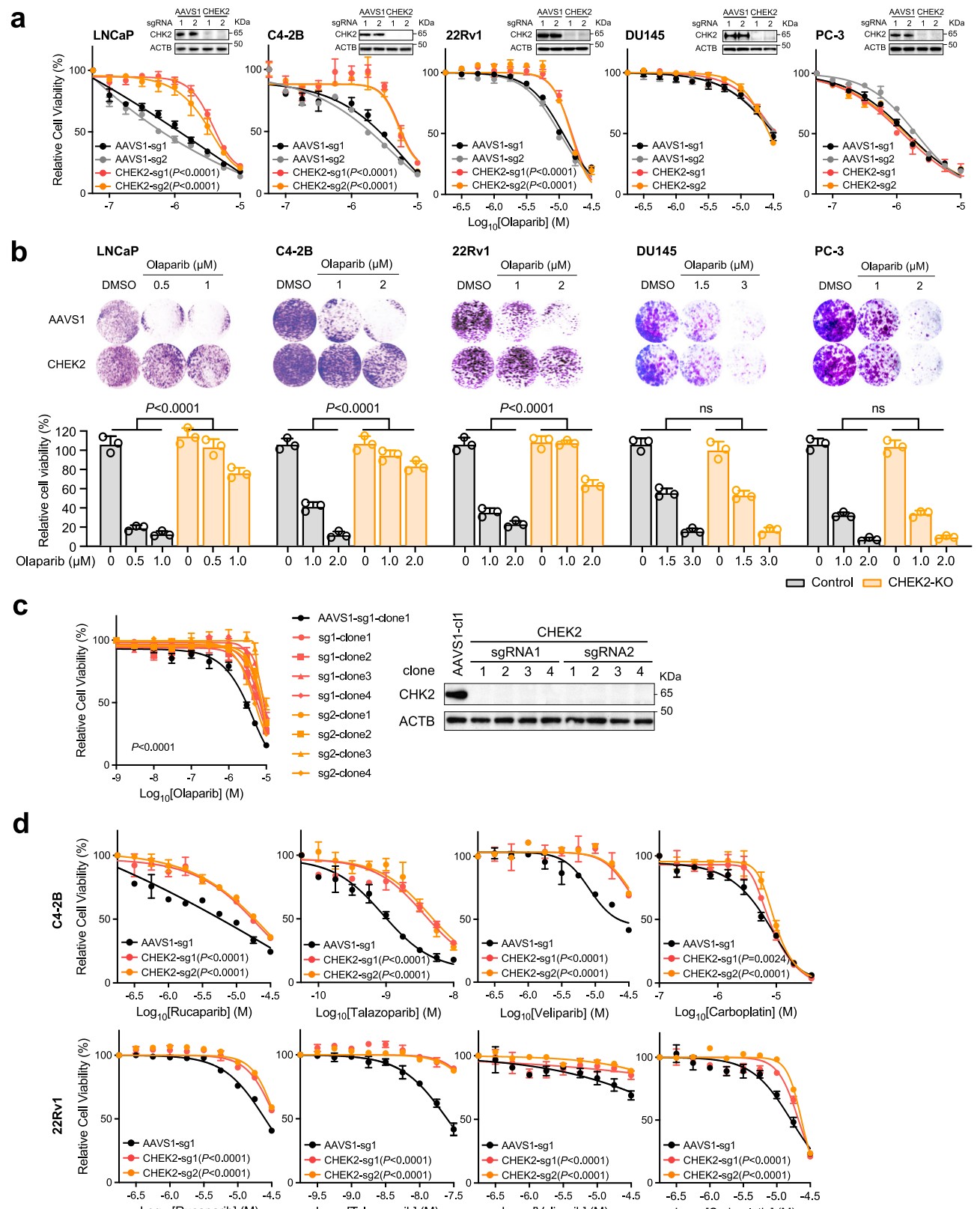

ATR inhibitor (ATRi)[65]. M6620 was remarkably more effective when combined with olaparib in cell viability assays (Fig. 8a). The combination therapy profoundly suppressed cell colony formation of *CHEK2*-KO cells in a synergistic manner, as demonstrated by the HSA and Bliss synergy scores[66,67] (Fig. 8b). Using in vivo xenograft models, we found that *CHEK2*-KO C4-2B and 22Rv1 tumors did not respond to olaparib or

M6620 as a single agent in contrast to *CHEK2*-intact control cells (Fig. 8c and Supplementary Fig 19). However, combination therapy abolished tumor growth. We did not observe any weight loss in each group, indicating the combination therapy was well tolerated. Mechanistically, M6620 inhibited CHK1 phosphorylation as expected (Fig. 8d). The combination treatment significantly inhibited BRCA1/2

**Fig. 6 | Loss of *CHEK2* renders PCa cells resistant to PARP inhibition. a** Dose-response curves after treatment with olaparib for two AAVS1 control (sg1 and sg2) and two *CHEK2*-KO (sg1 and sg2) LNCaP, C4-2B, 22Rv1, DU145, and PC-3 cell lines. The upper right panel in each cell line is the immunoblot analysis showing the CHK2 protein level in *CHEK2*-KO versus control cells. **b** Representative colony growth images (upper panel) and quantification (lower panel) after treatment with olaparib in AAVS1 control and *CHEK2*-KO PCa cell lines as indicated. Data are presented as mean ± SD of three biologically independent replicates. The *p*-values were determined using one-way ANOVA. **c** Dose-response curves (left panel) after treatment

with olaparib for AAVS1 control and *CHEK2*-KO C4-2B cell clones. Immunoblot analysis (right panel) showing the CHK2 protein level in *CHEK2*-KO and control cell clones. **d** Dose-response curves after treatment with rucaparib, talazoparib, veliparib, and carboplatin for AAVS1 control and *CHEK2*-KO C4-2B and 22Rv1 cells. In **a**, **c**, and **d** data are presented as mean ± SD (*n* = 3 biologically independent experiments). The immunoblot analyses were repeated independently twice with similar results. The *p*-values were determined by comparing *CHEK2*-KO to AAVS1 control cells using two-way ANOVA. Source data are provided as a Source Data file.

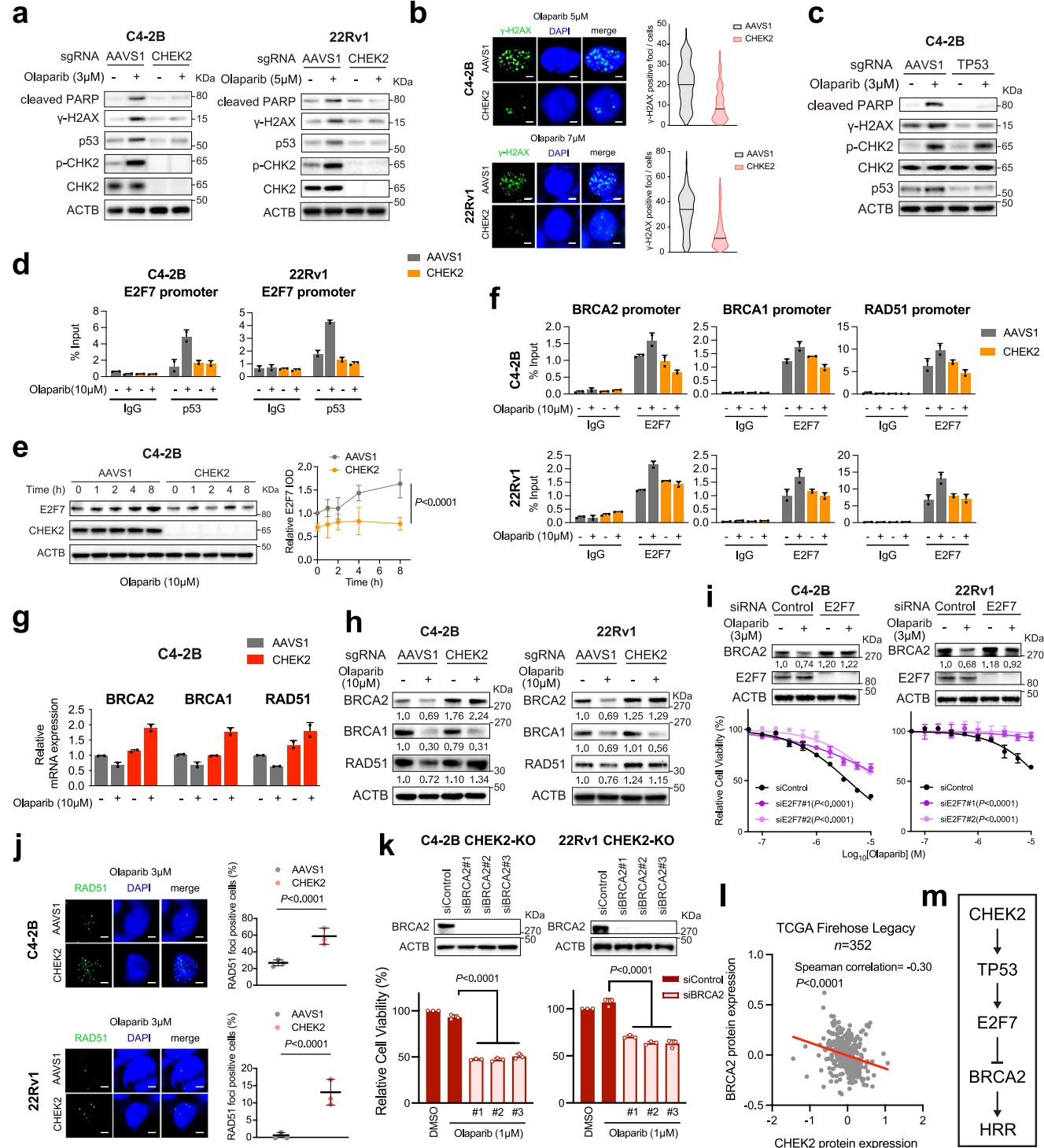

**Fig. 7 | Loss of *CHEK2*/*TP53* enhances HRR function through E2F7-controlled BRCA2 expression. a** Immunoblot analysis of the indicated proteins in AAVS1 control and *CHEK2*-KO C4-2B and 22Rv1 cells after treatment with and without olaparib for 72 h. The experiment was repeated independently three times with similar results. **b** Representative images of two biologically independent experiments and quantification of γ-H2AX foci in cells as described in (**a**) after treatment with olaparib for 24 h. More than 100 cells were analyzed per condition. Solid lines inside the violin indicate the median. The *p*-values were determined using two-sided *t*-test. Scale bar = 5 μm. **c** Immunoblot analysis of the indicated proteins in AAVS1 control and *TP53*-KO C4-2B cells after treatment with and without olaparib for 72 h. **d** p53 ChIP-qPCR at the *E2F7* promoter in AAVS1 control and *CHEK2*-KO C4-2B and 22Rv1 cells after treatment with or without olaparib for 24 h. **e** Immunoblot analysis (left panel) of E2F7 in AAVS1 control and *CHEK2*-KO C4-2B cells after treatment with olaparib for the indicated time. The expression of E2F7 is defined by ACTB normalized integrated optical density (IOD) (right panel). Data are presented as mean ± SD of three biologically independent experiments. **f** E2F7 ChIP-qPCR at the promoter regions of *BRCA1/2* and *RAD51* genes in cells as described in (**d**). **g** The mRNA expression of *BRCA1/2* and *RAD51* genes determined by RT-qPCR in cells as described in (**d**) after treatment with and without olaparib for 8 h. **h** Immunoblot analysis of the indicated proteins in cells as described in (**d**) after treatment with and without olaparib for 72 h. Normalized IOD values are indicated. **i** Immunoblot

analysis (upper panel) of BRCA2 in C4-2B and 22Rv1 cells after *E2F7* siRNA knockdown with or without olaparib treatment for 72 h. Dose-response curves (lower panel) after treatment with olaparib for C4-2B and 22Rv1 cells transfected with siRNAs against *E2F7* or control. Data are presented as mean ± SD (*n* = 3 biologically independent experiments). **j** Representative images and quantification of RAD51 foci in AAVS1 control and *CHEK2*-KO C4-2B and 22Rv1 cells after treatment with olaparib for 24 h. Data are presented as mean ± SD of three biologically independent replicates with more than 100 cells analyzed each replicate. Scale bar = 5 μm. **k** Cell viability after treatment with olaparib for *CHEK2*-KO C4-2B and 22Rv1 cells transfected with siRNAs against BRCA2 or control. Data are presented as mean ± SD (*n* = 3 biologically independent experiments). Immunoblot analysis of BRCA2 after siRNA knockdown (upper panel) is shown. **l** Scatter plot showing the correlation between CHK2 and BRCA2 protein levels in the TCGA cohort[25]. Spearman correlation coefficient and p-value are indicated. **m** Schematic model of HRR function regulated by the CHEK2-TP53-E2F7-BRCA2 pathway. In **a**, **c**, **h**, and **i**, the immunoblot analyses were repeated independently three times with similar results. In **d**, **f**, and **g**, data are presented as mean ± SD of two biologically independent experiments. In **e** and **i**, the *p*-values were determined using two-way ANOVA. In **j** and **k**, the *p*-values were determined using two-sided *t*-test. Source data are provided as a Source Data file.

and RAD51 expression in *CHEK2*-KO C4-2B and 22Rv1 cells, leading to more DNA damage and cell apoptosis as determined by γ-H2AX and cleaved-PARP expression. Functionally, olaparib-induced RAD51 formation was completely abolished in *CHEK2*-KO cells when they were treated in combination with M6620 (Fig. 8e).

## Discussion

In this study, we provided a systematic view of the genetic determinants and mechanisms underlying PARPi sensitivity and resistance in PCa beyond *BRCA1/2* alterations. We identified genes, such as *RAD51B*, *RAD54L*, and *FANCL*, that have already been used as biomarkers for PARP inhibition for mCRPC patients, as well as genes that are not in the FDA-approved genetic test. These genes identified in this study could serve as predictive biomarkers for PARP inhibition if mutated or deleted in PCa or therapeutic targets through pharmacologic inhibition in combination with PARPis. One such hit is *MMS22L*, a gene located in a genomic region frequently deleted in PCa, the loss of which induces BRCA-like response to PARPis. Mechanistically, MMS22L forms a complex with TONSL and accumulates at distressed replication forks, which is required for the HRR of replication fork-associated DSBs through promoting RAD51 loading[40,41,68]. When *MMS22L* is deleted, cells fail to load RAD51 to PARPi-induced collapsed replication forks, leading to the accumulation of DSBs, cell cycle arrest at G2/M phase, and apoptotic cell death. The HRR function of MMS22L-TONSL seems specific to the recovery of collapsed replication forks and does not act through BRCA2[40]. Interestingly, *MMS22L* deletion appears to be an early event in prostate tumorigenesis since they are more frequently detected in primary tumors. If the use of PARPis in localized PCa (e.g., in the neoadjuvant/adjuvant setting) is found to confer clinical benefits as has been suggested in breast cancer with *BRCA* mutations[69], detection of *MMS22L* deletion would identify a much larger population of patients who may benefit. In addition, our data suggests that cells with partial loss of *MMS22L* (i.e., monoallelic loss) are equally sensitive to PARP inhibition compared to complete loss (i.e., biallelic loss) likely due to insufficient MMS22L-TONSL complex formation. This is further supported by the evidence from clinical sample analyses, showing significantly lower mRNA levels and higher HRD scores in both homozygous and heterozygous *MMS22L* deletion tumors. Identification of *MMS22L* as a BRCAness gene explains a significant part of patients that have high HRD scores but do not have HRR gene alterations per the olaparib label. Given the considerable number of PCa patients with *MMS22L* deletion detected by next-generation sequencing- or DNA in situ hybridization-based assays, this

genomic alteration may be a valuable biomarker for PARP inhibition despite the fact that the response is *TP53*-dependent.

Previous studies of PARPi resistance have been focused on acquired resistance mechanisms in BRCA1/2-deficient cancers, showing that residual or restored HRR activity is the most commonly observed resistance mechanism. This can be achieved by secondary or reversion mutations of *BRCA1*, BRCA2, and *RAD51* isoforms[70–72], loss of *BRCA1* promoter methylation[73], increased expression of hypomorphic isoforms of *BRCA1*[74,75], loss of *TP53BP1* and resection-associated factors *RIF1*, *REV7*, and *Shieldin*[51,76–78]. In contrast, our work uncovers an intrinsic resistance mechanism in BRCA1/2-sufficient PCa cells, involving two frequently mutated tumor suppressor genes *TP53* and *CHEK2*. The role of p53 and its upstream activator CHK2 in a set of tightly regulated cell cycle checkpoints and DDR events has been extensively studied. In response to DNA DSBs, CHK2 is activated and phosphorylates and stabilizes p53, leading to cell cycle arrest to allow for DNA repair or inducing apoptosis after genotoxic damage. However, the PARPi resistance caused by *CHEK2*/*TP53* loss cannot be fully explained by impaired p53-mediated apoptosis. Indeed, PARPi can trigger apoptotic cell death in *TP53*-deficient PCa cells[79]. High-grade serous ovarian cancer (HGSOC) patients with defects in HRR are highly sensitive to PARP inhibition despite the fact that *TP53* mutations are detected in 96% of HGSOC tumors[80]. Here, we propose an additional pathway that contributes to PARPi resistance in PCa cells. We show that PARP inhibition activates the CHEK2-TP53-E2F7 pathway that suppresses the expression of HRR genes and potentiates the cytotoxicity of PARPis. Loss of *CHEK2* or *TP53*, however, markedly reduces the expression of their downstream target E2F7, leading to increased HRR gene (largely BRCA2) expression due to lack of E2F7-mediated transcriptional suppression. Upregulation of BRCA2 enhances HRR capacity sufficient for the repair of PARPi-induced DSBs and cell survival. Interestingly, BRCA1 is another direct target of the CHK2 kinase. The finding that loss of *CHEK2* leads to PARPi resistance instead of sensitivity is unexpected. We show that loss of *CHEK2* may compromise BRCA1 regulation; after all, the BRCA1 protein expression and its function are not completely abolished. On the other hand, the CHEK2-TP53-E2F7 pathway has predominantly emerged after PARP inhibition. While these results from preclinical models are still subject to clinical validation, our finding is consistent with the data from recent clinical trials which show little benefit in patients with mutations in *CHEK2*[8,10,81], providing a rationale to revisit FDA-approved clinically used biomarkers for the use of olaparib. Our data supports the notion that HRR deficiency resulting from alterations in non-*BRCA* genes is unlikely to

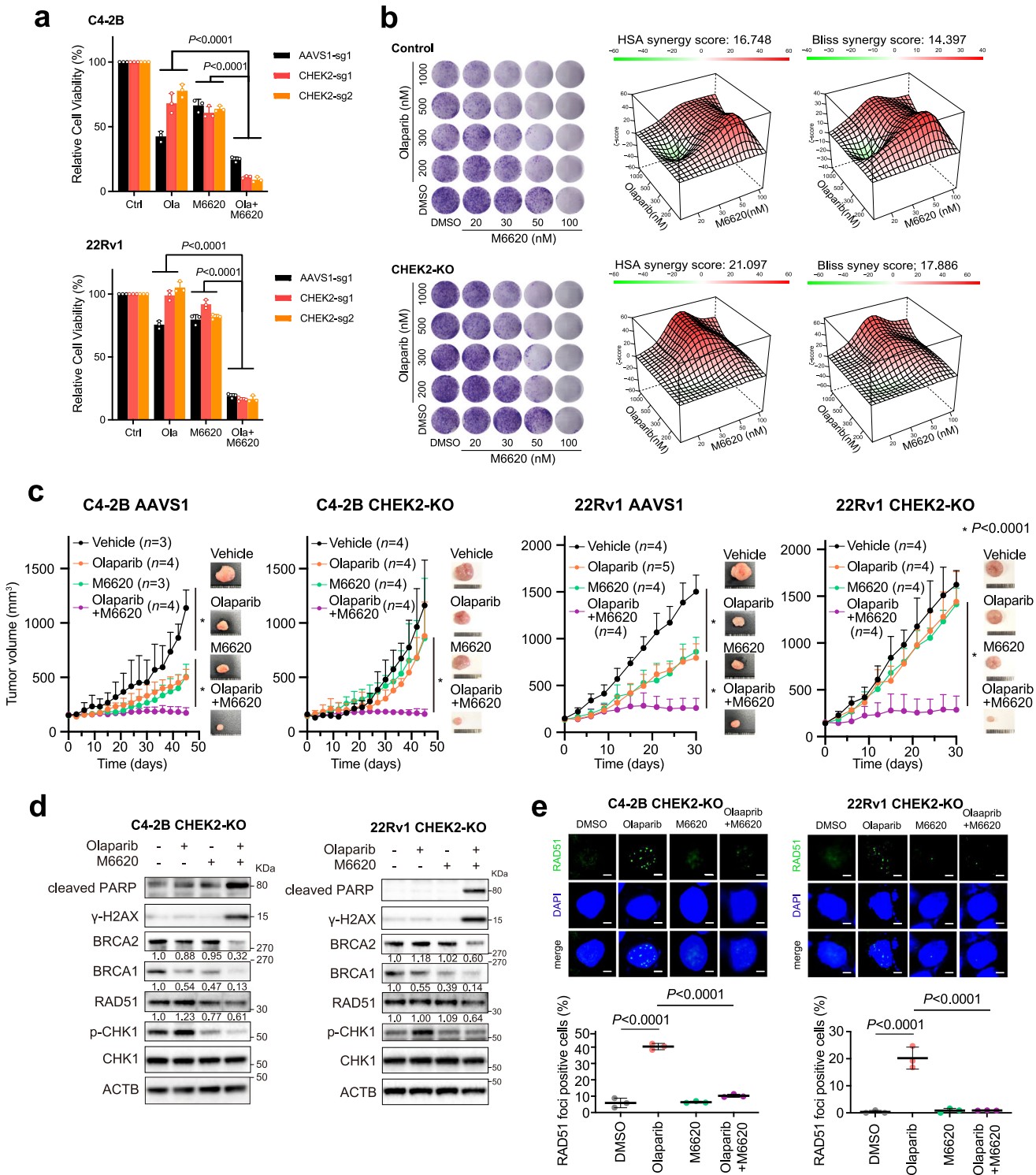

be of similar therapeutic relevance in comparison to BRCA2 deficiency, considering that mutations in the *TP53* gene occur in more than 50% of all cancers. Conversely, PCa harboring BRCA2 deficiency more likely responds to PARP inhibition regardless of TP53 status. This is consistent with our CRISPR screen results, showing that *BRCA2* was identified as one of the top hits in *TP53*-mutated DU145 cells, while many of the canonical HRR genes (including Fanconi anemia genes) were not negatively selected. Further investigation is needed to determine which genomic alterations in the HRR pathway can or cannot be rescued by upregulation of BRCA2 after loss of *CHEK2* or *TP5*3. We recently reported that RB1, another frequently mutated tumor

suppressor gene, may also confer resistance to PARP inhibition through E2F1-mediated upregulation of HRR genes[38], indicating the importance and complexity of E2F transcriptional network through activation and repression in the context of DNA repair. The finding of the TP53/E2F7- and RB1/E2F1-mediated resistance mechanisms is clinically relevant since *TP53* and *RB1* are concurrently altered in 39% of mCRPC tumors with adenocarcinoma histology and 74% of mCRPC tumors with neuroendocrine features[82,83]. Current clinical use of PARPis is guided by mutations of a HRR gene and overlooks common concurrent genomic alterations that may have an antagonistic effect.

**Fig. 8 | ATR inhibition overcomes PARPi resistance caused by *CHEK2* loss in PCa cells. a** Cell viability of AAVS1 control and *CHEK2*-KO C4-2B and 22Rv1 cells after treatment with DMSO, olaparib (3 μM), M6620 (100 nM) or combination of olaparib and M6620. Data are presented as mean ± SD (*n* = 3 biologically independent experiments). The *p*-values were determined using one-way ANOVA. **b** Representative colony growth images (left panel) of AAVS1 control and *CHEK2*-KO C4-2B cells after combination treatment of olaparib and M6620 as indicated. 3D synergy maps (right panel) of HSA and Bliss scores between olaparib and M6620 are shown. The experiments were repeated independently twice with similar results. **c** Xenograft mouse models using AAVS1 control and *CHEK2*-KO C4-2B and 22Rv1 cells. After tumors developed to a volume of around 150 mm³, mice were randomized into 4 treatment groups as vehicle, olaparib (50 mg/kg), M6620 (25 mg/kg), and olaparib plus M6620. The single agent was administered 5 days/week, while the combination treatment was given 5 days/week for olaparib plus 4 days/week for M6620. Tumor growth is shown with representative images. Data are presented as mean ± SD (*n* = 3, 4, 3, 4 in each group for C4-2B AAVS1; *n* = 4, 5, 4, 4 in each group for 22Rv1 AAVS1; *n* = 4, 4, 4, 4 in each group for C4-2B *CHEK2*-KO; *n* = 4, 4, 4, 4 in each group for 22Rv1 *CHEK2*-KO). The p-values were determined using two-way ANOVA. Tumor growth of each individual mouse is presented in Supplementary Fig. 19. **d** Immunoblot analysis of the indicated proteins in *CHEK2*-KO C4-2B and 22Rv1 cells after treatment with DMSO, olaparib (3 μM) and/or M6620 (100 nM) for 72 h. The normalized IOD values of the indicated proteins are shown. The experiments were repeated independently three times with similar results. **e** Representative images (upper panel) and quantification (lower panel) of RAD51 foci in *CHEK2*-KO C4-2B and 22Rv1 cells after treatment with DMSO, olaparib (10 μM) and/or M6620 (200 nM) for 24 h. Dots indicate each replicate with more than 100 cells analyzed. Data are presented as mean ± SD of three biologically independent experiments. The *p*-values were determined using two-sided *t*-test. Scale bar = 5 μm. Source data are provided as a Source Data file.

Finally, we propose a therapeutic approach with combined PARP and ATR inhibition to overcome PARPi resistance. In order to survive PARP inhibition, cancer cells rely more on ATR checkpoint to slow down cell cycle and reduce replication stress for repairing DSBs. ATR inhibition increases replication origin firing and DNA synthesis despite the presence of PARPi-induced replication fork gaps, leading to the progressive accumulation of DSBs[20]. Moreover, ATR inhibition can disrupt RAD51 loading to stalled replication forks and DSBs in PARPi-resistant BRCA1-deficient cells[84]. Our results further demonstrate that blocking ATR activity inhibits HRR gene expression and RAD51 foci formation, leading to a high number of stalled replication forks collapsing into toxic, irreparable DSBs. Indeed, the combination of PARPi with ATRi was synergistic in many PARPi-resistant models regardless of their resistance mechanisms[64]. Our previous studies support the use of combined PARP and ATR inhibition for PARPi-resistant prostate tumors with *RB1* loss[85]. Simultaneous suppression of ATR and PARP may be a preferred strategy to overcome PARPi resistance.

## Methods

### Cell lines and materials
Prostate cancer cell lines LNCaP, C4-2B, 22Rv1, PC-3, DU145, MDA PCa 2b, and UWB1.289 were obtained from American Type Culture collection (ATCC). Cells were cultured following the provider's recommendations. The AR-expressing and GFP-expressing PC-3 stable cell lines were obtained from Dr. Baruch Frenkel, University of Southern California, Los Angeles, CA[86]. The identity of the cell lines was confirmed based on high-resolution small tandem repeats (STR) profiling at Dana-Farber Cancer Institute (DFCI) Molecular Diagnostics Core Laboratory. Exome-sequencing analyses of PCa cell lines were performed by LC Sciences. Cells were routinely tested for mycoplasma. All reagents (including antibodies, small molecule inhibitors, oligonucleotides, and plasmids) used in this study are listed in Supplementary Data 7.

### Cell viability assay
PCa cells were seeded in 96-well plates at a density of 1000 cells per well for PC-3 and DU145, 1500 cells per well for C4-2B, 2000 cells per well for LNCaP, and 3000 cells per well for 22Rv1 cells in 100 μl 5% serum-containing media overnight. The adhered cells were treated with vehicle or inhibitors as indicated for 5–10 days. The media was replaced with fresh media with or without inhibitors every two days. Cell viability assays were performed using alamarBlue (ThermoFisher Scientific) according to the manufacturer's instruction. A dose-response curve was used to assess drug response.

### Clonogenic assay
PCa cells were seeded in 12-well plates at low density to avoid contact between clones. Subsequently, cells were treated as indicated and allowed to grow for 10–14 days. Colonies were fixed with paraformaldehyde (4%) for 20 min and stained with crystal violet (0.1%) for 15 min. Colony images were quantified using ImageJ software (National Institutes of Health).

### Cell growth competition assay
*MMS22L*-KO and AAVS1 control cells were infected with lentiviruses generated using LV-GFP (Addgene #25999) or LV-RFP (Addgene #26001) plasmid, respectively. *MMS22L*-KO LNCaP and C4-2B cells stably express GFP, whereas control cells stably express RFP. GFP or RFP positive cells were collected by the BD FACSAria cell sorter (BD Biosciences) and mixed in a 1:1 ratio. The mixed cell populations were incubated for 7 days with 3 uM of olaparib or DMSO treatment, followed by collection and calculation of GFP and RFP positive cells through flow cytometry.

### Drug synergy analysis
Cells were seeded in 12-well plates and treated with vehicle and 4 doses of olaparib (200, 300, 500, and 1000 nM) and M6620 (20, 30, 50, and 100 nM) in a matrix format to include 25 different dose combinations. After 14 days, colonies were fixed with paraformaldehyde (4%) for 20 min and stained with crystal violet (0.1%) for 15 min. Colonies were quantified using imageJ. Drug synergy scores were calculated based on the HSA and Bliss model using the SynergyFinder2.0[66]. A web-based tool (https://synergyfinder.fimm.fi) was used to determine synergistic drug combinations.

### RNA interference
*E2F7*, *TP53* and *BRCA2* siRNAs and negative control siRNA were purchased from Sigma-Aldrich. Cells were seeded onto 6-well or 96-well plates for 24 h and followed by siRNA (10 nM) transfection using Lipofectamine RNAiMAX (Invitrogen) according to the manufacturer's protocol. The siRNA sequences are listed in Supplementary Data 7.

### Cell cycle analysis
Cells were plated in 6-cm dishes with or without olaparib treatment for 72 h. Collected cells were labelled using Click-iT® EdU Flow Cytometry Assay Kit (Invitrogen) according to the manufacturer's protocol and followed by flow cytometry using BD LSR Fortessa (BD Biosciences). The gating strategy is provided in Supplementary Fig. 6. Cell cycle was analyzed using FlowJo Software (Vestion10.7.1).

### Chromatin immunoprecipitation quantitative PCR (ChIP-qPCR)
ChIP experiments were performed as previously described[87,88]. Briefly, PCa cells were grown in regular media and treated with olaparib (10 μM) for 24 h prior to ChIP. Cells were cross-linked by formaldehyde (1%) at room temperature (RT) for 10 min. After washing with ice-cold PBS, cells were collected and lysed. The soluble chromatin was purified and fragmented by sonication. Immunoprecipitation (IP) was performed using normal IgG or antibodies against p53 or E2F7 (2 μg/IP). ChIP DNA was extracted and analyzed by qPCR using iTaq Universal

SYBR Green Supermix (Bio-Rad). The primer sequences are listed in Supplementary Data 7. Each experiment was repeated 2 times.

## Quantitative RT-PCR (RT-qPCR)

After the indicated treatment, total RNA was extracted from cells using RNeasy Mini Kit (Qiagen) according to the manufacturer's protocol. RT-qPCR was performed as previously described[89]. Briefly, cDNA was prepared through reverse transcription using the iScript cDNA Synthesis Kit (Bio-Rad), and qPCR was conducted using SYBR Green PCR Master Mix (Applied Biosystems). Triplicate PCR reactions were conducted. *GAPDH* mRNA expression was analyzed for each sample in parallel. The primer sequences are listed in Supplementary Data 7.

## RNA sequencing (RNA-seq)

LNCaP and C4-2B cells were treated with olaparib (3 µM) or DMSO in two biological duplicates. Total RNAs were extracted using RNeasy Mini Kit (Qiagen). Quality of the extracted RNA was assessed by Agilent Bioanalyzer 2100 (Agilent). RNA-Seq library preparation and next-generation sequencing by Illumina HiSeq were conducted through GENEWIZ services. Raw RNA-seq reads are aligned to the human genome version hg38 by using STAR aligner[90]. Gene counts are quantified using HT-seq[91] with REFSEQ annotation. Differentially expressed genes are identified by using DESeq2[92] with cutoff of FDR < 0.01, and ranked based on the statistics. The GSEA software was used for determining the KEGG pathways[93].

## In vivo xenograft studies

All mice were maintained in compliance with the guidelines approved by the Institutional Animal Care and Use Committee (IACUC) at the Brigham and Women's Hospital. A 12 h light/12 h dark cycle is used. Temperatures of 65–75°F with 40–60% humidity are used. Every effort has been made to make sure that the animals do not suffer undue discomfort or distress. Humane euthanasia with carbon dioxide overdoes is applied as needed. For xenograft studies, male ICR/SCID mice and male NCG mice (NOD-*Prkdc*[em26CdS2]*Il2rg*[em26Cd22]/NjuCrl) at 4-5 weeks of age were purchased from Tconic Bioscience and Charles River Laboratories, respectively. For the PC-3 xenograft assay, 2.5 million cells were prepared in 50 µl PBS and mixed with 50 µl Matrigel (Corning Matrigel Matrix High Concentration #354262) to form a total of 100 µl cell suspension and followed by the subcutaneous inoculation in male ICR/SCID mice. Tumor volume was calculated using the modified ellipsoid formula: length x width$^2$/2. After tumors developed to a volume of approximately 150 mm$^3$, mice were randomized into 4 groups (vehicle: $n = 9$, doxycycline: $n = 5$, olaparib: $n = 5$, and doxycycline plus olaparib: $n = 5$). Mice were treated with vehicle (5% DMSO/10% d-a-tocopherol polyethylene glycol 1000 succinate; oral gavage, OG), doxycycline (10 mg/kg in 10% d-a-tocopherol polyethylene glycol 1000 succinate; intraperitoneal, IP), olaparib (50 mg/kg in 5%DMSO/10% d-a-tocopherol polyethylene glycol 1000 succinate; OG) or doxycycline (IP) plus olaparib (OG). Treatment was administered every day for doxycycline and 5 days a week for vehicle and olaparib.

For the C4-2B and 22Rv1 xenograft assays, 5 million cells of C4-2B or 2.5 million cells of 22Rv1 were prepared as above. C4-2B and 22Rv1 cells were injected subcutaneously into NCG and ICR/SCID mice, respectively. After tumors developed to a volume of approximately 150 mm$^3$, mice were randomized into 4 groups (vehicle, olaparib, M6620, and combination treatment) with 3–5 mice in each group as indicated. Mice were treated with vehicle, olaparib, M6620 (25 mg/kg in 5%DMSO/10% d-a-tocopherol polyethylene glycol 1000 succinate) or combination treatment (olaparib 50 mg/kg and M6620 25 mg/kg in 5%DMSO/10% d-a-tocopherol polyethylene glycol 1000 succinate) through OG. Treatment was administered 5 days a week for single agent, and 5 days a week of olaparib plus 4 days a week of M6620 for combination treatment. Tumor volume was manually measured every

3 days. The allowed maximal tumor size is 2 cm in any direction based on the institutional tumor production policies.

## Immunoblot analysis

Cells were lysed in RIPA buffer supplemented with Halt™ protease and phosphatase inhibitor cocktail (ThermoFisher Scientific). The protein concentration was determined using Pierce BCA Protein Assay Kit (ThermoFisher Scientific). Proteins were resolved in SDS-polyacrylamide gels (4–12%) and transferred to PVDF membranes using a Tris-glycine buffer system. Membranes were blocked with 5% non-fat milk in 0.1% Tween20 in TBS (TBS-T) for 1 h at room temperature, followed by incubation with primary antibodies and subsequently corresponding secondary antibodies in 5% milk TBS-T. The membranes were developed with Immobilon substrate (EMD Millipore). Immunoblot bands were quantified by integrated optical density (IOD) using ImageJ. Each protein IOD value was normalized by ACTB IOD. Antibodies used in immunoblot analysis are listed in Supplementary Data 7.

## Immunofluorescence analysis

Cells were grown in a poly-L lysine-coated 4 well Millicell EZ slides (EMD Millipore) for 48 h and pre-extracted with 0.2% Triton-X on ice 90 s before fixation with 4% paraformaldehyde in PBS for 20 min at room temperature (RT) and then permeabilized by incubation with ice-cold methanol. After permeabilization, cells were blocked with 5% BSA in PBS for 1 h at RT, followed by incubation with 5% BSA in TBS-T containing primary antibodies at a ratio of 1:200 overnight at 4 °C. Cells were washed and incubated with secondary fluorescent antibodies for 1 h at RT. Alexa Fluor 488 anti-rabbit and Alexa Fluor 488 anti-mouse antibodies were purchased from Invitrogen. After washing, the nuclear content was stained with Mounting Medium with DAPI (Abcam) overnight at 4 °C. Images were obtained with fluorescence microscope BX53 (Olympus). Images were quantitatively assessed using ImageJ software with 'Find maxima' function. More than 100 cells were analyzed per condition in each experiment in a blinded manner. Antibodies used in immunofluorescence analysis are listed in Supplementary Data 7.

## Generating knockout cell lines using CRISPR/Cas9 gene editing

CRISPR guides targeting each gene were cloned into lentiGuide-Puro vector (#52963; Addgene). The sgRNA sequences are listed in Supplementary Data 7. The lentiCas9-Blast vector that expresses Cas9 was obtained from Addgene (#52962). Lentiviruses were generated using packaging vectors pMD2.G (#12259; Addgene) and psPAX2 (#12260; Addgene) with Lipofectamine™ 3000 transfection reagents (#L3000015; Invitrogen) in 293FT cells. PCa cells were infected with lentiviruses expressing Cas9 and selected with Blasticidin (10 µg/ml) to establish stable Cas9-expressing cell lines. Polybrene was added at a final concentration of 8 ug/ml to increase infecting efficiency. To generate KO cells, PCa cells were infected with lentiviruses containing specific sgRNA and selected with puromycin (3 µg/ml). KO efficacy was determined by immunoblot analysis. Single-cell clones of *MMS22L*-KO and *CHEK2*-KO were generated using BD FACSAria cell sorter (BD Biosciences) and followed by expansion.

## Generation of TET-inducible exogenous MMS22L- or p53-expressing cells

*MMS22L*-KO and control C4-2B cells were infected with viruses containing TET-inducible *MMS22L* gene. The *MMS22L* cDNA was created to be resistant to *MMS22L* sgRNA1 by the introduction of silent mutations in the crispr RNA (crRNA) recognition sequence (change CTTGGCAGGAATATAGCACAA to CTAGGTAGAAATATAGCACAA). After neomycin (500 mg/ml) selection, doxycycline (0.15 µg/ml) induced MMS22L expression was confirmed by immunoblot. In addition, *MMS22L*-KO PC-3 cells were infected with viruses containing TET-inducible WT *TP53*.

Neomycin selection and the confirmation of p53 expression were performed as above. The *MMS22L* and *TP53* cDNAs were cloned into the Lenti-TRE3G-ORF-IRES-tRFP-PGK-Tet3G-neo vector. The plasmids were custom-designed and synthesized at Transomic Technologies.

## Genome-wide CRISPR-Cas9 knockout screen

We used genome-wide CRISPR-Cas9 KO H1 and H2 libraries obtained from Drs. Myles Brown and X. Shirley Liu's laboratories[15], consisting of over 180,000 sgRNAs (10 sgRNAs per gene). For the CRISPR-Cas9 KO screens, 200 million PCa cells were infected with the lentiviral CRISPR-Cas9 KO H1 and H2 libraries at a low multiplicity of infection (~0.3) to maximize the number of cells that have only one sgRNA integration. After 5 days of puromycin selection, the surviving cells were expanded. Following the preparation of 60 million cells for each condition to achieve a representation of at least 300 cells per sgRNA, cells were divided into day 0 control cells and cells cultured for 28 days (nine passages) treated with DMSO or olaparib before genomic DNA extraction and library preparation. We used olaparib at the concentration of 5 μM for LNCaP and C4-2B cells and 10 μM for 22Rv1 and DU145 cells. These concentrations were close to the half maximal inhibitory concentration (IC50) for each cell line and allowed us to identify both negatively (depleted) and positively (enriched) selected sgRNAs corresponding to gene knockouts that increase and decrease olaparib response, respectively. PCR was performed using genomic DNA to construct the sequencing libraries. Each library was sequenced at 30–40 million reads to achieve ~300 x average coverage over the CRISPR library. The library from day 0 sample of each screen served as controls to identify positively or negatively selected genes.

## CRISPR-Cas9 KO screen data analysis

CRISPR-Cas9 KO screen data was analyzed using the MAGeCK and MAGeCK-VISPR algorithms[94,95]. MAGeCK calculated the read counts for each sgRNA. MAGeCK-VISPR calculated the β-score for each gene by using AAVS1 gene as control. A comparison of the differential β-score between olaparib treatment and DMSO treatment was performed using MAGeCKFlute[17,18]. We ranked genes by differential β-score and robustly estimated σ, which is the standard deviation of the differential β-score by a "quantile matching" approach. A cut-off value was set to correspond 99% of the data falling within 2σ, which defined genes with lower β-scores than minus σ and higher β-scores than σ as negatively and positively selected genes, respectively. Identified genes were further analyzed using STRING protein interaction analysis[21]. GO analysis was performed using DAVID bioinformatics resources (https://david.ncifcrf.gov).

## DNAscope assay using tissue microarray (TMA)

The TMA was constructed using primary prostate tumors retrieved from the Vancouver Prostate Centre Tissue Bank as previously reported as approved by the Clinical Research Ethics Board[96]. Written informed consent was obtained from all subjects. A total of 146 tissue cores from 73 patients who had undergone radical prostatectomy were included in this study. The DNAscope assay (Advanced Cell Diagnostics, Newark, CA) is a chromogenic DNA in situ hybridization assay using two sets of target-specific probes. Two custom-designed probes were synthesized by Advanced Cell Diagnostics. DS-Hs-MMS22L-C1 is the probe targeting the *MMS22L* gene 29194-49180 of hg38 DNA range = chr 6:97142161-97283437. DS-Hs-CEP6p-C2 is the chromosome enumeration control probe targeting the centromeric region 20891-35730 of hg38 DNA range = chr 6: 57224489-57312704. The assay was performed on the TMA using DNAscope™ HD Duplex Reagent Kit (Advanced Cell Diagnostics) according to the manufacturer's protocol. Briefly, the tissue section was pretreated to allow access to target DNAs and followed by hybridization with two sets of probes. Two independent signal amplification systems were used to detect both target DNAs. Probes were hybridized to a cascade of signal amplification molecules, culminating in

binding of enzyme-labeled probes. Two chromogenic substrates were used, and probe-targeted regions were visualized in red for the *MMS22L* locus and blue for the control region (i.e., the centromeric region of chromosome 6p). An increase in the number of red dots relative to blue dots indicates a copy number gain or amplification, while a decrease in the number of red dots or no red dots indicates a copy number loss or deletion. The images were evaluated and quantified visually by a pathologist (L. Fazli, Vancouver Prostate Centre). The *MMS22L* wild-type was defined as the ratio of the total number of red dots divided by the total number of blue dots greater than or equal to 0.5 (red/blue ≥ 0.5). The heterozygous deletion was defined as the ratio of red/blue between 0.1–0.5, while the homozygous deletion was defined as the ratio of red/ blue less than or equal to 0.1 (red/blue ≤ 0.1).

## HRD and HRDetect scores

HRD and HRdetect scores were determined as previously described[46]. A total of 267 PCa cases with whole genome sequencing data were analyzed. HRD score was defined as the cumulative sum of loss-of-heterozygosity (LOH), large scale transitions (LST), and number of telomeric allelic imbalances (ntAI) and determined using the scarHRD R package[97]. HRDetect score was defined by six distinguishing mutational signatures predictive of *BRCA1/2* deficiency and determined using HRDetect algorithm[47].

## Statistical analysis

Statistical analyses were performed using the unpaired two-sided Student's *t* test, one-way or two-way ANOVA with a post hoc Tukey's honest significant difference (HSD) test when comparing at least three conditions using the Prism software (GraphPad). *P*-values of less than 0.05 were considered as statistically significant.

## Public data analysis

Clinical datasets were analyzed using the cBio Portal for Cancer Genomics (cBioPortal; www.cbioportal.org) and PCaDB[98] (http://bioinfo.jialab-ucr.org/PCaDB/). Integrated Genome Viewer (https://www.broadinstitute.org/igv/) was used for visualization of ChIP-seq data. ChIP-seq data were obtained from ChIP-Atlas[99] (http://chip-atlas.org).

## Data availability

Source data are provided with this paper. The publicly available PCa clinical data used in this study, including gene expression and genomic alteration, are available in the cBioPortal database[25,26] (www.cbioportal.org) and PCaDB[98] (http://bioinfo.jialab-ucr.org/PCaDB/). The publicly available ChIP-seq data used in this study are available in the GEO database and the ChIP-Atlas database under accession code: GSM2671296, GSM3378513, GSM2296278, GSM545807, GSM501692, GSM1366696, GSM2988952, GSM981236, GSM991661, GSM2825525, GSM1208730[62,100–109] (http://chip-atlas.org)[99]. RNA-seq data generated in this study are available under GEO accession code GSE189186. The remaining data are available within the Article, Supplementary Information or Source Data file. Source data are provided with this paper.

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

## Acknowledgements

This work was supported by NCI 1R21CA267496 (to L.J.) and DoD W81XWH-22-1-0477 (to L.J.). We thank Max Greenberg for financial support, in honor of his parents, Leon and Esther Greenberg. We thank Myles Brown, X. Shirley Liu, Xiaoqing Wang, and Zach Herbert for helping in CRISPR screening. We thank S. Kinoshita, Z. D. Nagel, C. G. Piett, T. Tian, S. Jia, and J. Geng for constructive discussion and technical support.

## Author contributions

T.Tsujino and T.Takai performed the majority of experiments and analyses. T.Tsujino, T.Takai, and K.H undertook the CRISPR screens. T.Tsujino and L.J. wrote the manuscript and prepared figures. F.G. performed animal studies. T.Tsutsumi, X.B., C.M, C.F., B.G., and A.S. performed cell studies and provided technical assistance. Z.Sztupinszki performed HRD analyses. N.X., L.F. and X.D. performed DNAscope assays. H.A. and A.S.K. provided administrative support. T.Tsujino, T.Takai, K.H, A.D.C., K.W.M., Z. Szallasi, L.Z., A.S.K., L.J. analyzed the data and revised the manuscript. L.J. conceptualized, designed the study, and acquired funding for the present study.

## Competing interests

The authors declare no competing interests.
