## [Peer Review File · Nature Communications]

Reviewers' Comments:

Reviewer #1:

Remarks to the Author:

Aggressive cancer cells including prostate cancer (PCa) harbor mutations in DNA repair machinery to create further genomic instability, which leads to gene mutations. Not surprisingly, genes involved in DNA damage response and repair (DDR), especially homologous recombination repair (HRR) genes, are among the most frequently altered in aggressive and advanced tumors. This is an interesting paper that deals with a timely subject, i.e., on PARPi sensitivity and resistance in PCa. A significant % of patient prostate tumors suffer BRCA1/2 mutations and/or deficiencies/alterations in other DDR including HRR genes, and at least 10 phase II/III clinical trials of treating mCRPC patients with PPARis are finished or ongoing. For example, in May 2020, USFDA approved Olaparib (PARP1 inhibitor) for patients with mCRPC carrying germline BRCA1/2 alterations after the PROfound clinical trial.

On the other hand, how the non-BRCA DDR deficient prostate tumors may respond to PPARi remains much less clear. Thus, the current preclinical studies in 4 BRCA1/2-proficient PCa cell lines (LNCaP, C4-2B, 22Rv1 and DU145) have significant clinical implications. By conducting CRISPR-Cas9 dropout (KO) screening, the team identified both negatively and positively selected genes potentially involved in mediating the sensitivity vs. resistance, respectively, of PCa cells to Olaparib. For example, varying numbers of 'positively selected genes' were identified in the 4 cell types after 28 days of Olaparib exposure, which represent the genes enriched in Olaparib-surviving cells and are thus potentially involved in mediating PARPi resistance. Overall, their results imply that Olaparib treatment may not be restricted to mCRPC patients with BRCAness but alterations in other HRR genes may also render the tumors susceptible to PARP inhibition. Hence, this study may have profound impact in the field and extend the horizon of Olaparib treatment beyond BRCA1/2.

Major points:

1. Authors stated that "These four cell lines and have no deleterious biallelic mutations in BRCA1/2 and other canonical HRR genes" but Supplementary Table 1 showed that LNCaP cells have homozygous missense mutations in ATM, BRCA2, CDK12, CHEK1 and TP53. Other cell types also seemed to have homozygous mutations in DDR and HRD genes. This seemed to represent a discordance. Also, all Supplementary Tables should have a Table legend (or heading).
2. In their CRISPR screen (Fig. 1), authors relied on genes shared by at least two cell lines. Given that C4-2B is a subline of LNCaP, not surprisingly most of the hits were shared between these two cell lines. This consideration somewhat reduces the enthusiasm for the results.
3. As shown in Fig. 2, there were only two HRR genes, HELLS and WDR76, that were lost in all four tested cell lines. Surprisingly, authors didn't follow up on these two HRR genes but instead chose to study MMS22L without providing a strong rationale. This is important because MMS22L loss would also introduce synthetic lethality like BRCAness but needs a functional p53, which is frequently deleted or mutated in mCRPC and especially NEPC.
4. Intriguingly, MMS22L or TONSL loss did not sensitize p53-mut DU145 and p53-null PC3 cells (Supplementary Fig. 7) to Olaparib (Fig. 3a). Although authors linked this insensitivity to p53 functional deficiency (Fig. 5a), this reviewer wonders whether this may also implicate a potential AR/AR signaling requirement in PPARi sensitization by MMS22L/TONSL loss (especially when considering that AR signaling itself has been associated with DNA damage). It's true that the 3 cell lines, 22Rv1, DU145 and PC3 all have p53 mutations/deletion and manifest low Olaparib sensitivity (Supplementary Fig. 1), but both DU145 and PC3 are also AR-negative and 22Rv1 cells express extremely low levels of AR.
5. Analysis of RAD51 recruitment to DSBs (Fig. 4 related) may not be enough to conclude that HRR is impaired or functional. Authors should consider performing γ -H2AX/RAD51 co-staining and, importantly, analyzing HRR and NHEJ efficiency directly in their experiments using HRR and NHEJ reporters as described in several papers (Nat Methods. 2011. 8(8):671-6). As the cell lines chosen in this study are claimed to be all BRCA1/2 proficient, authors should use these reporters to analyze the effect of MMS22L loss on HRR activity because restoration of MMS22L would lead to RAD51 recruitment at the damage sites. Authors also ignored NHEJ pathway entirely. In the absence of HRR, the NHEJ repair is the go-to DDR pathway.
6. The observations that CHEK2 loss confers PCa cell resistance (rather than sensitivity) to

Olaparib is indeed surprising as noted by authors and need more explanation. CHEK2 upon activation phosphorylates BRCA1, so technically without CHEK2 BRCA1 is not getting activated and the cells should be more susceptible to PARPi. To clarify, authors may analyze the ATM and ATR pathways in their experiments to figure out the interplay between these two related pathways.

7. Fig. 7. Why were the levels of γ -H2AX lower in CHEK2/p53 KO cells following Olaparib treatment (Fig. 7a and c)? These genes are downstream of DSBs and knocking out these genes should not have any effect on γ -H2AX levels or should perhaps increase the levels. Also, as p-CHK2 is a direct readout of ATM activity in response to DSBs, why were the levels of p-CHK2 protein similar in Olaparib treated AAVS1 and TP53 KO cells even though there were not much DSBs in the TP53-KO group (Fig. 7c)? Finally, as shown in Fig. 7j, CHEK2 KO led to increased RAD51 foci as compared to AAVS1 KO in both C4-2B and 22RV1 cells. This is surprising as RAD51 protein also gets recruited at the DSBs and as shown in Fig. 7a/b, CHEK2 KO actually decreased the γ -H2AX levels in cells following Olaparib treatment. Authors should co-stain γ -H2AX and RAD51 to clarify.

8. Fig. 8: This last comment also applied to Figure 8e also and authors should co-stain γ -H2AX and RAD51 in these cells. In Figure 8a, in M6620-treated group, with both CHK1 and CHK2 non-functional cells were not dying, which is a bit surprising as cancer cells always undergo some DNA damage because of replication stress or mitotic defects because of their high proliferation rate. Can authors analyze levels of γ -H2AX as well in these groups.

Minor points:

1. In their CRISPR screen, it is not clear whether the loss of genes that conferred PARPi sensitivity is due to cell death and/or growth inhibition.
2. In Fig. 5d, how come the control AAVS1 siMMS22L lines did not start at 100%?
3. The percentage mentioned on page 10, line 237 does not match the values in Fig 5j.
4. There are some typos in the text and figure legends. For example, Fig. 1l on page 7 (line 156) should be Fig. 1i.

Reviewer #2:

Remarks to the Author:

In this study, the authors deployed a CRISPR KO library screen to identify genetic modulators of prostate cancer cell's sensitivity to the PARP inhibitor (PARPi) olaparib. Olaparib was approved by the FDA for castration resistant, metastatic prostate cancers that harbor certain mutations in the DNA damage HR pathway including BRCA1/2, ATM, CHK2 and others. Identifying genes whose deletion can sensitize BRCA WT cells towards olaparib could broaden the biomarker space for olaparib in prostate cancer. Towards this end, the authors identified MMS22L, a known DNA damage repair gene, whose deletion results in olaparib sensitivity in prostate cancer cells in a p53-dependent manner. The authors showed that cells with MMS22L deletion had defective homologous repair, and olaparib killing requires WT p53 function. These findings suggests that prostate cancer patients whose tumors harbor MMS22L deletion but retain WT p53 function could potentially benefit from Olaparib treatment. This is an exciting discovery that has immediate translational potential.

The second part of the manuscript dealt with resistance mechanism to olaparib in BRCA WT cells. This part is a bit confusing because BRCA WT cells are generally insensitive to, due to their HR proficiency. Thus, it would make more sense to carry out the resistance screen in a BRCA mutant cell line instead. The authors made the discovery that CHK2 deletion made cells more resistant to olaparib. This is an opposite finding to the established literature. The authors showed that in CHK2 KO cells, BRCA2 was upregulated, which may account for the resistance mechanism. This could be an adaptation response to CHK2 KO in these cells. Resistance to olaparib can be overcome by combined PARP and ART inhibition. The value of the second half of the story is limited in its current form, given CHK2 mutation is a biomarker for olaparib in prostate cancer patients. If the authors could provide evidence that elevated BRCA2 expression is a clinically relevant resistance mechanism in CHK2 mutant tumors treated with olaparib, it would make this part of the story stronger.

Specific points:

1. Figure 2B and Figure S5: How does the acute deletion of MMS22L affect baseline cell viability and cell cycle? MMS22L is required for DNA repair due to fork collapse and its deletion has been reported to be quite lethal in cells.
2. In LNCaP or C4-2B cells, does p53 knockdown decrease g-H2AX foci in MMS22L KO cells?
3. How does CHK2 KO and E2F7 KO affect base line cell viability relative to control sgRNA/siRNAs in the absence of drug treatment?
4. Figure 6. Does CHK2 inhibitors also make these cells resistant to olaparib? The authors' finding is a surprising one and contradicts the existing literature. Thus it would be useful to clarify whether this is an acute effect or an adaptive response in the cells to CHK2 KO.
5. Figure 8c and 8d: How do parental cells (i.e. CHK2 WT) respond to drug inhibition in the xenograft models, are they equally sensitive to the combination?

Reviewer #3:

Remarks to the Author:

This study by Tsujino et al represents a significant step forward in understanding the potential clinical utility of PARP inhibitors in prostate cancer, albeit all their findings are subject to clinical validation which is not part of the current manuscript.

Overall, this is a very good manuscript, with experiments well planned; the data is presented in a clear and comprehensive manner.

There are some issues which I think need to be acknowledged as in diverse passages of the paper, the authors make quite strong claims which in the opinion of this reviewer should be tempered:

- First of all, the authors note that BRCA1/2 did not come up as one of their top hits in the CRISPR experiment for putative PARPi sensitizing alterations. This has been described before and, as the authors acknowledge, it may relate to these events being too toxic for the cells per se. This makes me wonder if the same may happen for other events such as PALB2, and if, as result, the events detected may be in reality a "second tier" of alterations in terms of clinical actionability.
- Could the authors clarify why Figure 2B only present C4-2 as a model? Would it not make sense use the same models as for the CRISPR screening?
- Figure 3 and lines 161-162- "We showed that deletion of either MMS22L or TONSL resulted in significantly increased sensitivity to olaparib in LNCaP, C4-2B, MDAPCa2b and 22Rv1 cells, but not in DU145 or PC-3 cells." – the effect in 22Rv1 seems very modest to be honest; any explanation of why the impact of MMS22L would be different between models? If anything, the lesser effect in castration-resistant models goes against using MMS22L as predictive biomarker for PARPi in this setting...
- Could the authors explain the difference between Figure 4e and Suppl Fig 6? Both present different HRD scar measurements. In the one presented in Suppl Fig 6, the association with MMS22L loss does not seem that strong as in Figure 4 (not sure what the * means in Suppl Fig 6).
- In the same Figure 4E, the main finding is that 4-5 of the 11 cases with MMS22L hom-del cases cluster close to the BRCA2 deficient ones. Considering the small numbers, can the authors check if there are any HRR alterations in these cases? How does this HRD score relate to their further finding of MMS22L loss only sensitizing TP53 WT tumors to PARPi? Are these 4-5 cases in the Figure 4e TP53 WT or mutated?
- Lines 261-262: We deleted CHEK2 in five PCa cell lines and observed significantly increased resistance in LNCaP, C4-2B, 22Rv1 cells, but not in TP53-mutant PC-3 and DU145 cells using cell viability and colony formation assays – I am struggling to interpret this finding considering all these cell line models are, by themselves, resistant to PARPi. What does "increase of resistance" mean in an already resistant model? Could the authors elaborate the discussion? The latter part of the paper all emanates from this concept so it is relevant.
- The authors conclude that concomitant ATR and PARP inhibition overcomes CHEK2-mediated resistance. But the authors have previously published in ATR and PARP inhibition being synergistic in prostate cancer models. Is the effect different/more potent in these CHEK2-deficient models? In Figure 8a it does not look like it is.

We thank reviewers for their insightful comments on our manuscript. We have tried our best to address every one of them. Below we provide a point-by-point response to the reviewers' concerns. Reviewer comments are shown in **Bold**. We have highlighted the changes within the manuscript in red.

Reviewer #1:

Major points:

1. Authors stated that “These four cell lines and have no deleterious biallelic mutations in BRCA1/2 and other canonical HRR genes” but Supplementary Table 1 showed that LNCaP cells have homozygous missense mutations in ATM, BRCA2, CDK12, CHEK1 and TP53. Other cell types also seemed to have homozygous mutations in DDR and HRD genes. This seemed to represent a discordance. Also, all Supplementary Tables should have a Table legend (or heading).

We agree that we didn't have a definition of “deleterious mutations” in Supplementary Table 1. We have now added a table legend, in which we defined deleterious mutations as “high-impact” mutations determined by Ensemble Variant Effect Predictor (VEP) or “pathogenic” and “likely pathogenic” mutations in CinVar. We marked these mutations in Supplementary Table 1.

It should be noted that the interpretation of genetic variants is not clear-cut. Here, we used VEP to determine the degree of their effects on gene function. And most missense mutations identified in our cell lines have little impact on protein production or protein function. They are considered as benign even though some of them are biallelic.

2. In their CRISPR screen (Fig. 1), authors relied on genes shared by at least two cell lines. Given that C4-2B is a subline of LNCaP, not surprisingly most of the hits were shared between these two cell lines. This consideration somewhat reduces the enthusiasm for the results.

We agree that there are more common genes identified between C4-2B and LNCaP cells since they are derived from the same tumor. On the other hand, studies including ours have shown major differences in genetic background (including mutation number and distribution) and gene expression profiles between LNCaP and C4-2B cells (Decker et al., 2012; Spans et al., 2014). Although we have tried different approaches to define common hits, our conclusion remains unchanged. Importantly, the findings of this study do not fully rely on how we define the common hits. In fact, we have analyzed both common and unique genes for each cell line and provided a global view of genetic determinants in PARP inhibition. Following the global analyses, we selected MMS22L and CHEK2/TP53 for further mechanistic investigation of PARP inhibitor sensitivity and resistance respectively. This is largely based on their high impact on PARP inhibition and frequent alterations in prostate cancer.

References:

Decker, K.F., Zheng, D., He, Y., Bowman, T., Edwards, J.R., and Jia, L. (2012). Persistent androgen receptor-mediated transcription in castration-resistant prostate cancer under androgen-deprived conditions. *Nucleic Acids Res* 40, 10765-10779. 10.1093/nar/gks888.

Spans, L., Helsen, C., Clinckemalie, L., Van den Broeck, T., Prekovic, S., Joniau, S., Lerut, E., and Claessens, F. (2014). Comparative genomic and transcriptomic analyses of LNCaP and C4-2B prostate cancer cell lines. *PLoS One* 9, e90002. 10.1371/journal.pone.0090002.

3. As shown in Fig. 2, there were only two HRR genes, HELLS and WDR76, that were lost in all four tested cell lines. Surprisingly, authors didn't follow up on these two HRR genes but instead chose to study MMS22L without providing a strong rationale. This is important because MMS22L loss would also introduce synthetic lethality like BRCAness but needs a functional p53, which is frequently deleted or mutated in mCRPC and especially NEPC.

Yes. HELLS and WDR76 are the only two negatively selected genes identified in all four cell lines, which is very interesting. However, they are not a top priority in this study largely because they are not the most frequently altered genes in prostate cancer. It should be noted that there are many interesting genes (including HELLS and WDR76) identified from our CRISPR screens, which are currently under investigation. We didn't include these data in this manuscript because studies are still superficial, and many efforts are needed to mechanistically elucidate the roles of these genes in PARP inhibition.

In this study, we have focused on one of frequently deleted genes MMS22L (up to 14%). Given the considerable number of prostate tumors with MMS22L deletion, this genomic alteration is a valuable biomarker for PARP inhibition despite the fact that the response is TP53-dependent. Furthermore, our data suggest that cells with MMS22L heterozygous deletion are equally sensitive to PARP inhibition compared to homozygous deletion. This is further supported by the evidence from clinical sample analyses, showing significantly lower mRNA levels and higher HRD scores in both homozygous and heterozygous MMS22L deletion tumors. These findings may expand the population of patients to benefit from PARP inhibition beyond BRCA1/2 mutation carriers.

We previously argued that current targeted cancer therapies are largely guided by mutations of a single gene, which overlooks concurrent genomic alterations (Miao et al., 2022). We showed that PARP inhibitor sensitivity is influenced by RB1 loss through the E2F1 pathway. Here, we further demonstrated that TP53 status may also impact PARP inhibition through the E2F7 pathway. We are encouraged by these findings. Indeed, our results may explain some of the disparate clinical results from clinical trials due to interaction between multiple genomic alterations and support a comprehensive genomic test to determine who may benefit from PARP inhibition.

References:

Miao, C., Tsujino, T., Takai, T., Gui, F., Tsutsumi, T., Sztupinszki, Z., Wang, Z., Azuma, H., Szallasi, Z., Mouw, K.W., et al. (2022). RB1 loss overrides PARP inhibitor sensitivity driven by RNASEH2B loss in prostate cancer. *Sci Adv* 8, eab19794. 10.1126/sciadv.ab19794.

4. Intriguingly, MMS22L or TONSL loss did not sensitize p53-mut DU145 and p53-null PC3 cells (Supplementary Fig. 7) to Olaparib (Fig. 3a). Although authors linked this insensitivity to p53 functional deficiency (Fig. 5a), this reviewer wonders whether this may also implicate a potential AR/AR signaling requirement in PPARi sensitization by

MMS22L/TONSL loss (especially when considering that AR signaling itself has been associated with DNA damage). It's true that the 3 cell lines, 22Rv1, DU145 and PC3 all have p53 mutations/deletion and manifest low Olaparib sensitivity (Supplementary Fig. 1), but both DU145 and PC3 are also AR-negative and 22Rv1 cells express extremely low levels of AR.

It is well known that the crosstalk between AR signaling pathway and DNA repair pathway occurs at multiple levels. On the one hand, AR regulates DNA repair gene expression (Polkinghorn et al., 2013) and AR pathway inhibitor induces “BRCAness” (Li et al., 2017), which forms the basis for several clinical trials using anti-androgens (enzalutamide or abiraterone) in combination with PARP inhibitors (olaparib, talazoparib, or niraparib) for metastatic CRPC. Initial results from the PROpel and the MAGNITUDE trials are inconsistent (presented at ASCO GU 2022). It remains controversy whether the combination can go beyond BRCA mutation carriers. On the other hand, studies including ours have also demonstrated that PARP1 and PARP2 are involved in AR-mediated transcription and function as a co-factor (Gui et al., 2019; Schiewer et al., 2012). Thus, unlike p53, prostate cancer cells with functional AR are likely less sensitive to PARP inhibition.

We have examined PARP inhibitor response in AR-positive MMS22L-KO C4-2B cells after blocking AR signaling with enzalutamide (ENZ). As expected, AR inhibition further sensitizes MMS22L-KO cells to Olaparib (see the figure below), supporting the combination therapy with antiandrogen and PARP inhibitor. Notably, the effect of enzalutamide is relatively small because C4-2B cells are already very sensitive to olaparib after MMS22L deletion. We didn't include this data in the manuscript since it's irrelevant to the study.

MMS22L-KO cells respond to the anti-AR and anti-PARP combination therapy. MMS22L-KO C4-2B cells were treated with AR antagonist enzalutamide (ENZ) (1 uM or 5uM) in combination with Olaparib as indicated for 5 days. Cell viability was measured by the AlamarBlue assay. The p-values were determined by comparing DMSO to ENZ treatment using one-way ANOVA.

To address the reviewer's concern, we restored AR expression in AR-negative PC-3 cells and examined cell response to olaparib after MMS22L knockdown. Unlike p53, AR restoration did not sensitize PC-3 cells to olaparib. We have included the data in new Supplementary Fig 9, supporting that insensitivity of MMS22L-KO PC-3 cells to PARP inhibition is likely due to lack of p53, but not AR.

References:

- Polkinghorn, W.R., Parker, J.S., Lee, M.X., Kass, E.M., Spratt, D.E., Iaquina, P.J., Arora, V.K., Yen, W.F., Cai, L., Zheng, D., et al. (2013). Androgen receptor signaling regulates DNA repair in prostate cancers. *Cancer Discov* 3, 1245-1253. 10.1158/2159-8290.CD-13-0172.
- Li, L., Karanika, S., Yang, G., Wang, J., Park, S., Broom, B.M., Manyam, G.C., Wu, W., Luo, Y., Basourakos, S., et al. (2017). Androgen receptor inhibitor-induced "BRCAness" and PARP inhibition are synthetically lethal for castration-resistant prostate cancer. *Sci Signal* 10. 10.1126/scisignal.aam7479.
- Gui, B., Gui, F., Takai, T., Feng, C., Bai, X., Fazli, L., Dong, X., Liu, S., Zhang, X., Zhang, W., et al. (2019). Selective targeting of PARP-2 inhibits androgen receptor signaling and prostate cancer growth through disruption of FOXA1 function. *Proc Natl Acad Sci U S A* 116, 14573-14582. 10.1073/pnas.1908547116.
- Schiewer, M.J., Goodwin, J.F., Han, S., Brenner, J.C., Augello, M.A., Dean, J.L., Liu, F., Planck, J.L., Ravindranathan, P., Chinnaiyan, A.M., et al. (2012). Dual roles of PARP-1 promote cancer growth and progression. *Cancer Discov* 2, 1134-1149. 10.1158/2159-8290.CD-12-0120.

5. Analysis of RAD51 recruitment to DSBs (Fig. 4 related) may not be enough to conclude that HRR is impaired or functional. Authors should consider performing γ -H2AX/RAD51 co-staining and, importantly, analyzing HRR and NHEJ efficiency directly in their experiments using HRR and NHEJ reporters as described in several papers (Nat Methods. 2011. 8(8):671-6). As the cell lines chosen in this study are claimed to be all BRCA1/2 proficient, authors should use these reporters to analyze the effect of MMS22L loss on HRR activity because restoration of MMS22L would lead to RAD51 recruitment at the damage sites. Authors also ignored NHEJ pathway entirely. In the absence of HRR, the NHEJ repair is the go-to DDR pathway.

First, the role of MMS22L in homologous recombination (HR) has been well defined. Previous studies have revealed that MMS22L and TONSL are required for the HR-mediated repair of replication fork-associated double strand breaks (DSBs) (Duro et al., 2010; O'Donnell et al., 2010). The MMS22/TONSL complex is specifically located at the stalled or collapsed replication forks. This function is conserved in yeast and mammalian cells. Studies have also shown that Mms22 is essential for HR induced by agents that block replication forks, but not for HR-mediated repair of radiation- or endonuclease-induced DSBs (Duro et al., 2008). Therefore, the HR reporter assay (using I-SceI endonuclease) may not be an appropriate approach to determine the effect of MMS22L loss on replication fork-associated HR activity.

Second, we have performed DNA repair reporter assays in collaboration with Dr. Zachary D. Nagel's lab (Nagel et al., 2014) to measure HR and NHEJ repair capacity after MMS22L/TONSL loss. We observed a minor reduction in HR repair and no change in NHEJ repair after MMS22L or TONSL knockout. In contrast, over 50% reduction of HR was observed in BRCA1-null UWB1 ovarian cancer cells. We didn't include this data in the manuscript because we think the reporter assay does not reflect the true function of the MMS22L/TONSL complex as previously defined. Instead, we focused on the effect of MMS22L loss on RAD51 loading. Further investigation is needed to understand the different roles of MMS22L vs. BRCA1/2 in HR, which is beyond the scope of this manuscript.

Finally, there are three commonly used approaches to predict PARP inhibitor response in clinical practice. They are: 1) deleterious mutations in HRR genes (genetic biomarkers); 2) RAD51 foci formation (functional biomarker) (Castroviejo-Bermejo et al., 2018); 3) HRD score (genomic scar-based biomarker). In this study, we have shown that the loss of MMS22L significantly impairs RAD51 foci formation, a central step of HRR. We also detected increased HRD score in tumors with MMS22L deletion. Other than BRCA1/2, high HRD score is not commonly observed in prostate tumors with mutations in HRR genes (such as BLM, PALB2, RAD50, RAD51, RAD52, RAD54B, and RAD54L) (Sztupinszki et al., 2020). Together, our results suggest that loss of MMS22L confer a BRCA-like synthetic lethality to PARP inhibition.

References:

- Duro, E., Lundin, C., Ask, K., Sanchez-Pulido, L., MacArtney, T.J., Toth, R., Ponting, C.P., Groth, A., Helleday, T., and Rouse, J. (2010). Identification of the MMS22L-TONSL complex that promotes homologous recombination. *Mol Cell* 40, 632-644. 10.1016/j.molcel.2010.10.023.
- O'Donnell, L., Panier, S., Wildenhain, J., Tkach, J.M., Al-Hakim, A., Landry, M.C., Escribano-Diaz, C., Szilard, R.K., Young, J.T., Munro, M., et al. (2010). The MMS22L-TONSL complex mediates recovery from replication stress and homologous recombination. *Mol Cell* 40, 619-631. 10.1016/j.molcel.2010.10.024.
- Duro, E., Vaisica, J.A., Brown, G.W., and Rouse, J. (2008). Budding yeast Mms22 and Mms1 regulate homologous recombination induced by replisome blockage. *DNA Repair (Amst)* 7, 811-818. 10.1016/j.dnarep.2008.01.007.
- Nagel, Z.D., Margulies, C.M., Chaim, I.A., McRee, S.K., Mazzucato, P., Ahmad, A., Abo, R.P., Butty, V.L., Forget, A.L., and Samson, L.D. (2014). Multiplexed DNA repair assays for multiple lesions and multiple doses via transcription inhibition and transcriptional mutagenesis. *Proc Natl Acad Sci U S A* 111, E1823-1832. 10.1073/pnas.1401182111.
- Castroviejo-Bermejo, M., Cruz, C., Llop-Guevara, A., Gutierrez-Enriquez, S., Ducy, M., Ibrahim, Y.H., Gris-Oliver, A., Pellegrino, B., Bruna, A., Guzman, M., et al. (2018). A RAD51 assay feasible in routine tumor samples calls PARP inhibitor response beyond BRCA mutation. *EMBO Mol Med* 10. 10.15252/emmm.201809172.
- Sztupinszki, Z., Diossy, M., Krzystanek, M., Borcsok, J., Pomerantz, M.M., Tisza, V., Spisak, S., Ruzs, O., Csabai, I., Freedman, M.L., and Szallasi, Z. (2020). Detection of Molecular Signatures of Homologous Recombination Deficiency in Prostate Cancer with or without BRCA1/2 Mutations. *Clin Cancer Res* 26, 2673-2680. 10.1158/1078-0432.CCR-19-2135.

6. The observations that CHEK2 loss confers PCa cell resistance (rather than sensitivity) to Olaparib is indeed surprising as noted by authors and need more explanation. CHEK2 upon activation phosphorylates BRCA1, so technically without CHEK2 BRCA1 is not getting activated and the cells should be more susceptible to PPARis. To clarify, authors may analyze the ATM and ATR pathways in their experiments to figure out the interplay between these two related pathways.

We agree with the reviewer that CHEK2 phosphorylates and stabilizes BRCA1, which is the basis of CHEK2 being one of the FDA-approved biomarkers for olaparib. We were surprised as

well when we identified CHEK2 as one of the top hits in the positive selection causing resistance rather than sensitivity after deletion.

In this study, however, we provide strong evidence showing HRR genes (BRCA1/2 and RAD51) are the downstream targets of the CHEK2-TP53-E2F7-mediated transcriptional repression (Fig. 7d-g). Accordingly, we found that the mRNA levels of BRCA1/2 and RAD51 were significantly upregulated after CHEK2 deletion. On the protein levels, BRCA2 and RAD51 were also upregulated, while the BRCA1 protein level was decreased (Fig. 7h). This is likely because BRCA1 is directly modulated by CHEK2 through posttranslational modification despite transcriptional upregulation. Our data suggest that the enhanced HRR function in CHEK2-KO cells is largely due to upregulation of BRCA2 and RAD51. In terms of BRCA1, we think the BRCA1 function remains to some extent although its phosphorylation is somewhat impaired without CHEK2. We recently received NIH R21 funding to further elucidate the underlying mechanisms.

In clinical trials, it remains controversial whether prostate cancer patients harboring mutations in non-BRCA HRR genes truly benefit from PARP inhibition. Gene-by-gene analysis revealed that patients with BRCA1/2 mutations often exquisitely respond to PARP inhibition, other HRR genes are hard to be assessed because of small patient number. As for CHEK2 alterations, the result from the phase III PROfound (olaparib) trial showed no overall survival benefit in patients with CHEK2 mutations (n=12). In the phase II TRITON2 (rucaparib) trial, only 1 out of 9 patients had confirmed objective response. In the phase II TALAPRO-1 (talazoparib) trial, none of the patients (n=9) showed confirmed objective response. While several clinical trials are still ongoing, current clinical data do not support that patients with CHEK2 mutations would benefit from PARP inhibition.

Finally, in response to DNA damage, the ATM-CHK2 and ATR-CHK1 pathways are activated. The functions and interplay of these two pathways have been extensively studied. In this study, we have shown that olaparib treatment upregulates CHK2 phosphorylation in CHEK2-intact cells (Fig. 7a), indicating increased ATM activity. On the other hand, loss of CHEK2 activates ATR pathway demonstrated by increased CHK1 phosphorylation (Supplementary Fig. 17), supporting the interplay between these two pathways.

7. Fig. 7. Why were the levels of γ -H2AX lower in CHEK2/p53 KO cells following Olaparib treatment (Fig. 7a and c)? These genes are downstream of DSBs and knocking out these genes should not have any effect on γ -H2AX levels or should perhaps increase the levels. Also, as p-CHK2 is a direct readout of ATM activity in response to DSBs, why were the levels of p-CHK2 protein similar in Olaparib treated AAVS1 and TP53 KO cells even though there were not much DSBs in the TP53-KO group (Fig. 7c)? Finally, as shown in Fig. 7j, CHEK2 KO led to increased RAD51 foci as compared to AAVS1 KO in both C4-2B and 22Rv1 cells. This is surprising as RAD51 protein also gets recruited at the DSBs and as shown in Fig. 7a/b, CHEK2 KO actually decreased the γ -H2AX levels in cells following Olaparib treatment. Authors should co-stain γ -H2AX and RAD51 to clarify.

We have provided evidence demonstrating that CHEK2/TP53 KO leads to enhanced HRR capacity due to derepression of E2F7-mediated gene regulation. That is why we observed less DSBs (γ -H2AX foci) because they were repaired through HR (Fig. 7a and c). We also observed increased activity of ATM-CHK2 pathway (increased CHK2 phosphorylation) after olaparib

treatment. However, TP53 KO had no effect on CHK2 phosphorylation because p53 is a downstream effector of CHK2.

As the reviewer suggested, we performed co-staining of RAD51 and γ -H2AX in CHEK2-KO C4-2B and 22Rv1 cells in comparison to the control AAVS1 cells (new Supplementary Fig. 16a). We observed increased RAD51 but decreased γ -H2AX foci in the same cells after olaparib treatment, supporting enhanced HRR capacity after CHEK2 deletion and reduced DNA double strand breaks (DSBs).

8. Fig. 8: This last comment also applied to Figure 8e also and authors should co-stain γ -H2AX and RAD51 in these cells. In Figure 8a, in M6620-treated group, with both CHK1 and CHK2 non-functional cells were not dying, which is a bit surprising as cancer cells always undergo some DNA damage because of replication stress or mitotic defects because of their high proliferation rate. Can authors analyze levels of γ -H2AX as well in these groups.

We agree that cells cannot survive loss of both CHK1 and CHK2. In Figure 8e, RAD51 foci were measured within 24h before cells undergo apoptosis. On the other hand, we have detected significantly increased γ -H2AX (DSBs) and cleaved PARP (apoptosis) after the combination treatment for 3 days (Fig. 8d).

Minor points:

1. In their CRISPR screen, it is not clear whether the loss of genes that conferred PARPi sensitivity is due to cell death and/or growth inhibition.

It could be both. In general, PARP inhibitor (PARPi)-induced cell death is a relatively slow process. The p53 pathway is activated immediately after PARPi treatment leading to cell cycle arrest followed by cell apoptosis if DNA damage is overwhelming. We normally measure cell apoptosis after the treatment for 3 days. A recent study from Dr. Lee Zou's lab (co-authors) showed that PARPi induces double strand breaks (DSBs) progressively and HR-deficient cells fail to slow down and repair DSBs over multiple cell cycles (Simoneau et al., 2021). Since cells have undergone 9 passages over 28 days in the CRISPR screen, we believe the negative selection is caused by cell death. That said, we cannot rule out that some genes are not critical in HRR, loss of which may only slow down cell growth.

References:

Simoneau, A., Xiong, R., and Zou, L. (2021). The trans cell cycle effects of PARP inhibitors underlie their selectivity toward BRCA1/2-deficient cells. *Genes Dev* 35, 1271-1289. 10.1101/gad.348479.121.

2. In Fig. 5d, how come the control AAVS1 siMMS22L lines did not start at 100%?

We have corrected Fig. 5d, and all controls have now started at 100%.

3. The percentage mentioned on page 10, line 237 does not match the values in Fig 5j.

We thank the reviewer for pointing out our mistake. We have now corrected the number. The results of all cases can be found in Source Data.

4. There are some typos in the text and figure legends. For example, Fig. 1l on page 7 (line 156) should be Fig. 1i.

We have corrected typos throughout the manuscript.

Reviewer #2:

The second part of the manuscript dealt with resistance mechanism to olaparib in BRCA WT cells. This part is a bit confusing because BRCA WT cells are generally insensitive to, due to their HR proficiency. Thus, it would make more sense to carry out the resistance screen in a BRCA mutant cell line instead. The authors made the discovery that CHK2 deletion made cells more resistant to olaparib. This is an opposite finding to the established literature. The authors showed that in CHK2 KO cells, BRCA2 was upregulated, which may account for the resistance mechanism. This could be an adaptation response to CHK2 KO in these cells. Resistance to olaparib can be overcome by combined PARP and ART inhibition. The value of the second half of the story is limited in its current form, given CHK2 mutation is a biomarker for olaparib in prostate cancer patients. If the authors could provide evidence that elevated BRCA2 expression is a clinically relevant resistance mechanism in CHK2 mutant tumors treated with olaparib, it would make this part of the story stronger.

There is no clinical evidence because of limited patient number.

We agree with the reviewer that it would make more sense to carry out the resistance screen in a BRCA mutant cell line. *First*, our initial goal of carrying out CRISPR screens in four BRCA1/2-proficient cell lines is to expand the use of PARP inhibitors beyond BRCA1/2 mutations and find new synthetically lethal interactions. *Second*, the BRCA mutant prostate cancer cell lines were not available when we started the screens. We have now generated BRCA2-KO in a population of LNCaP, C4-2B, 22Rv1, and DU145 cells. We are still in a process of generating a single cell clone with a complete BRCA2 KO in these lines. *Third*, sensitivity or resistance to PARP inhibition is not black and white. As we have shown, C4-2B, and 22Rv1 cells are partially responsive to olaparib in vitro (Fig. 8a) or in vivo (Fig. 8c). Using these cell lines, we hope to identify genes, loss of which increases or decreases PARP inhibitor response in the same screen. Finally, we have identified many well-known resistance genes, including PARP1/2, PARG, ARH3, TP53BP1, and E2F7, indicating our screens are powerful. In comparison with published data from CRISPR screens using BRCA-deficient cells (Clements et al., 2020; Zimmermann et al., 2018), our results are fully comparable, if not better.

We agree that the identification of CHEK2 as one of the top hits from positive selection is surprising. The reviewer #1 raised the same question. We have addressed this concern in the question #6. Ultimately, our findings are subject to clinical validation with more patients.

References:

Clements, K.E., Schleicher, E.M., Thakar, T., Hale, A., Dhoonmoon, A., Tolman, N.J., Sharma, A., Liang, X., Imamura Kawasawa, Y., Nicolae, C.M., et al. (2020). Identification of regulators of poly-ADP-ribose polymerase inhibitor response through complementary CRISPR knockout and activation screens. *Nat Commun* 11, 6118. 10.1038/s41467-020-19961-w.

Zimmermann, M., Murina, O., Reijns, M.A.M., Agathangelou, A., Challis, R., Tarnauskaite, Z., Muir, M., Fluteau, A., Aregger, M., McEwan, A., et al. (2018). CRISPR screens identify genomic ribonucleotides as a source of PARP-trapping lesions. *Nature* 559, 285-289. 10.1038/s41586-018-0291-z.

Specific points:

1. Figure 2B and Figure S5: How does the acute deletion of MMS22L affect baseline cell viability and cell cycle? MMS22L is required for DNA repair due to fork collapse and its deletion has been reported to be quite lethal in cells.

We didn't observe significant changes in cell viability and cell cycle after MMS22L deletion by sgRNA knockout or siRNA knockdown. We agree that MMS22L is required for replication fork-associated HRR, but prostate cancer cells seem to tolerate MMS22L deletion. In fact, a large number of prostate tumors harbor MMS22L homozygous deletion (Fig. 2a).

Another example is the gene MCM6, which is the member of the MCM2-7 heterohexamer, a DNA helicase essential for DNA replication. Again, prostate cancer cells appear to survive MCM6 deletion (Fig. 2b). Homozygous MCM6 deletion was observed in 6% primary prostate tumors (Fig. 2a). It will be interesting to understand how prostate cancer cells can survive the loss of these essential genes in the future.

2. In LNCaP or C4-2B cells, does p53 knockdown decrease γ -H2AX foci in MMS22L KO cells?

We have performed the experiment as the reviewer suggested. In new supplementary Fig. 8, we have shown that TP53 knockdown significantly decreases γ -H2AX foci in MMS22L-KO C4-2B cells. Accordingly, cells become less sensitive to olaparib.

3. How does CHK2 KO and E2F7 KO affect base line cell viability relative to control sgRNA/siRNAs in the absence of drug treatment?

In new supplementary Fig. 13, we have shown that CHEK2 or TP53 knockout does not affect C4-2B and 22Rv1 cell viability and cell cycle. We didn't generate E2F7 knockout cell lines. Instead, we have shown that E2F7 siRNA knockdown has no effect on C4-2B and 22Rv1 cell viability either (new Supplementary Fig. 15).

4. Figure 6. Does CHK2 inhibitors also make these cells resistant to olaparib? The authors' finding is a surprising one and contradicts the existing literature. Thus it would be useful to clarify whether this is an acute effect or an adaptive response in the cells to CHK2 KO.

A recent study showed that genetic or pharmacological inhibition (using small molecule inhibitor BML277) of CHK2 blunted PARP inhibitor response in hematologic cells (Xu et al.,

2021). They found that blocking CHK2-mediated phosphorylation of p53 impaired olaparib response. These results are in line with ours.

To address the reviewer's concern, we also treated C4-2B and 22Rv1 cells with CHK2 inhibitor (BML277) and found little impact on olaparib response (see Figure below). It should be noted that pharmacological inhibition does not equal genetic deletion. We didn't include and discuss this result in the manuscript because it is still superficial. Further investigation is needed to understand acute CHK2 inhibition vs. long-term genetic deletion.

References:

Xu, Z., Vandenberg, C.J., Lieschke, E., Di Rago, L., Scott, C.L., and Majewski, I.J. (2021). CHK2 Inhibition Provides a Strategy to Suppress Hematologic Toxicity from PARP Inhibitors. *Mol Cancer Res* 19, 1350-1360. 10.1158/1541-7786.MCR-20-0791.

5. Figure 8c and 8d: How does parental cells (i.e. CHK2 WT) respond to drug inhibition in the xenograft models, are they equally sensitive to the combination?

We performed additional xenograft experiments using CHEK2-intact AAVS1 control C4-2B and 22Rv1 cells. We found that these tumors were equally sensitive to the combination treatment (see new Fig. 8c). Notably, CHEK2-intact tumors are partially sensitive to olaparib or M6620 as a single agent in contrast to CHEK2-KO cells, supporting that loss of CHEK2 promotes PARP inhibitor resistance.

Reviewer #3

• First of all, the authors note that BRCA1/2 did not come up as one of their top hits in the CRISPR experiment for putative PARPi sensitizing alterations. This has been described before and, as the authors acknowledge, it may relate to these events being too toxic for the cells per se. This makes me wonder if the same may happen for other events such as PALB2, and if, as result, the events detected may be in reality a “second tier” of alterations in terms of clinical actionability.

We agree with the reviewer that our findings are subject to clinical validation. In terms of PALB2, we examined raw read counts from our CRISPR screens. Like BRCA1/2, we found that sgRNAs targeting PALB2 were depleted with or without olaparib treatment, supporting PALB2 as a fitness gene in prostate cancer cells (see new Supplementary Fig 4.).

However, we disagree with the reviewer that the genes detected may be in a “second tier” of alterations. In fact, most of the canonical HRR genes were identified in our screens (see Fig. 1g-i and Table 2). BRCA1 and BRCA2 were identified as one of the top hits in C4-2B (ranked #27) and DU145 cells (ranked #3) respectively. They were highly ranked among all negative selection genes based on an average β -score from four cell lines (see Fig. 1i). In addition, we identified many novel genes (such as MMS22L and MCM6), which have been considered as essential genes. Prostate cancer cells seem to survive the loss of these genes. Compared to CRISPR screening in non-prostate cancer cells (Clements *et al.*, 2020; Olivieri *et al.*, 2020; Zimmermann *et al.*, 2018), our screening is equally powerful, if not better. We have addressed a similar question in Reviewer #2 question #1.

References:

Clements, K.E., Schleicher, E.M., Thakar, T., Hale, A., Dhoonmoon, A., Tolman, N.J., Sharma, A., Liang, X., Imamura Kawasawa, Y., Nicolae, C.M., *et al.* (2020). Identification of regulators of poly-ADP-ribose polymerase inhibitor response through complementary CRISPR knockout and activation screens. *Nat Commun* 11, 6118. 10.1038/s41467-020-19961-w.

Olivieri, M., Cho, T., Alvarez-Quilon, A., Li, K., Schellenberg, M.J., Zimmermann, M., Hustedt, N., Rossi, S.E., Adam, S., Melo, H., *et al.* (2020). A Genetic Map of the Response to DNA Damage in Human Cells. *Cell* 182, 481-496 e421. 10.1016/j.cell.2020.05.040.

Zimmermann, M., Murina, O., Reijns, M.A.M., Agathangelou, A., Challis, R., Tarnauskaite, Z., Muir, M., Fluteau, A., Aregger, M., McEwan, A., *et al.* (2018). CRISPR screens identify genomic ribonucleotides as a source of PARP-trapping lesions. *Nature* 559, 285-289. 10.1038/s41586-018-0291-z.

• Could the authors clarify why Figure 2B only present C4-2 as a model? Would it not make sense use the same models as for the CRISPR screening?

Our initial validation has been focused on genes frequently altered and related to DDR functions. The genes presented in Figure 2B happened to be the ones all identified in C4-2B/LNCAp cells. We then move on to study MMS22L and CHEK2 in different cell models. We agree that this is a weakness in this study. Indeed, we have validated genes in other cell lines as well, including some of unique genes (such as mitochondrial complex I genes) in DU145 cells. We have decided to publish those genes separately.

• Figure 3 and lines 161-162- “We showed that deletion of either MMS22L or TONSL resulted in significantly increased sensitivity to olaparib in LNCaP, C4-2B, MDAPCa2b and 22Rv1 cells, but not in DU145 or PC-3 cells.” – the effect in 22Rv1 seems very modest to be honest; any explanation of why the impact of MMS22L would be different between models? If anything, the lesser effect in castration-resistant models goes against using MMS22L as predictive biomarker for PARPi in this setting...

It is an accurate observation by the reviewer that 22Rv1 cells had a modest effect after MMS22L deletion. This can be explained by the fact that 22Rv1 cells harbor a monoallelic TP53 mutation which may influence p53 activity to some extent. The p53 protein level is relatively lower in 22Rv1 cells compared to TP53 WT cells (See Supplementary Fig 7). Yes. MMS22L deletion as a predictive biomarker is p53-dependent. This finding further supports the utility of a comprehensive genomic test instead of a single gene-based prediction in clinical practice.

• Could the authors explain the difference between Figure 4e and Suppl Fig 6? Both present different HRD scar measurements. In the one presented in Suppl Fig 6, the association with MMS22L loss does not seem that strong as in Figure 4 (not sure what the * means in Suppl Fig 6).

Both HRD and HRDetect scores are an analytic approach to measure genome instability. HRD score is an unweighted sum of three independent DNA-based measures of genomic instability (loss of heterozygosity, telomeric allelic imbalance, and large-scale transitions) in the tumor. The HRD score has been widely accepted to predict PARP inhibitor response. On the other hand, HRDetect score is a whole-genome sequencing (WGS)-based classifier designed to predict BRCA1/2 deficiency based on six mutational signatures (Davies et al., 2017). In general, the HRD score and HRDetect score show good correlation. Although redundant, a recent prostate cancer study from Drs. Szupinszki and Szallasi (co-authors) showed that HRDetect was more accurate identifying the BRCA-deficient cases (Sztupinszki *et al.*, 2020). Here, we used both approaches to demonstrate BRCA-like HRD after MMS22L loss.

References:

Davies, H., Glodzik, D., Morganella, S., Yates, L.R., Staaf, J., Zou, X., Ramakrishna, M., Martin, S., Boyault, S., Siewerts, A.M., et al. (2017). HRDetect is a predictor of BRCA1 and BRCA2 deficiency based on mutational signatures. *Nat Med* 23, 517-525. 10.1038/nm.4292.

Sztupinszki, Z., Diossy, M., Krzystanek, M., Borcsok, J., Pomerantz, M.M., Tisza, V., Spisak, S., Rusz, O., Csabai, I., Freedman, M.L., and Szallasi, Z. (2020). Detection of Molecular Signatures of Homologous Recombination Deficiency in Prostate Cancer with or without BRCA1/2 Mutations. *Clin Cancer Res* 26, 2673-2680. 10.1158/1078-0432.CCR-19-2135.

• In the same Figure 4E, the main finding is that 4-5 of the 11 cases with MMS22L homo-del cases cluster close to the BRCA2 deficient ones. Considering the small numbers, can the authors check if there are any HRR alterations in these cases? How does this HRD score relate to their further finding of MMS22L loss only sensitizing TP53 WT tumors to PARPi? Are these 4-5 cases in the Figure 4e TP53 WT or mutated?

We have re-examined MMS22L homo-deletion cases and found no other concurrent HRR alterations. It should be noted that high HRD score is not commonly observed in prostate tumors with mutations in HRR genes (other than BRCA1/2) (Sztupinszki *et al.*, 2020). Our results may partially explain previously unclarified cause of HRD in prostate cancer beyond BRCA1/2 mutations.

HRD score (a genomic scar analysis) offers a stable readout of a tumor's lifetime HRR competency, reporting a defect in HRR occurred at some point in tumorigenesis, but not at the point of treatment. A variety of mechanisms could restore HRR function after genomic scarring.

Indeed, tumors with BRCA1/2 mutations may also develop resistance to PARP inhibition due to loss of 53BP1 despite high HRD scores. During prostate cancer development and progression, MMS22L loss likely occurs before TP53 alteration. We have observed more frequent MMS22L deletion in primary tumors, while TP53 mutations are detected more in metastatic tumors. In addition, high HRD scores were observed in both TP53 WT and mutant tumors harboring MMS22L deletion. However, we agree that the number of cases in the cohort is too small to determine whether and to what extent TP53 impacts MMS22L-mediated HRD.

References:

Sztupinszki, Z., Diossy, M., Krzystanek, M., Borcsok, J., Pomerantz, M.M., Tisza, V., Spisak, S., Ruzs, O., Csabai, I., Freedman, M.L., and Szallasi, Z. (2020). Detection of Molecular Signatures of Homologous Recombination Deficiency in Prostate Cancer with or without BRCA1/2 Mutations. *Clin Cancer Res* 26, 2673-2680. 10.1158/1078-0432.CCR-19-2135.

• Lines 261-262: We deleted CHEK2 in five PCa cell lines and observed significantly increased resistance in LNCaP, C4-2B, 22Rv1 cells, but not in TP53-mutant PC-3 and DU145 cells using cell viability and colony formation assays – I am struggling to interpret this finding considering all these cell line models are, by themselves, resistant to PARPi. What does “increase of resistance” mean in an already resistant model? Could the authors elaborate the discussion? The latter part of the paper all emanates from this concept so it is relevant.

We agree that we didn't clearly define “sensitivity” and “resistance” in the manuscript because it's not all black and white. We have modified the text to avoid using “significantly increased resistance” when we quantify the response to PARP inhibitors. Instead, we used “significantly reduced olaparib response”.

• The authors conclude that concomitant ATR and PARP inhibition overcomes CHEK2-mediated resistance. But the authors have previously published in ATR and PARP inhibition being synergistic in prostate cancer models. Is the effect different/more potent in these CHEK2-deficient models? In Figure 8a it does not look like it is.

The synergistic effect of the combination therapy is measured based on the colony formation assay (Fig. 8b). Importantly, we have also shown that CHEK2-KO cells are less sensitive to olaparib or M6620 as a single agent compared to AAVS1 control cells in vitro (Fig. 8a) and in vivo (new Fig. 8c). When two drugs are combined, their overall effect is greater than the sum of their individual effects, supporting the synergistic effect of the combination therapy.

Reviewers' Comments:

Reviewer #1:

Remarks to the Author:

As various PARPi's are moving to the clinical arena to treat mCRPC patients, it's important to identify and expand the patient population that may benefit from PARPi's, and to identify potential efficacy and resistance biomarkers. In this context, the current manuscript is significant reporting that BRCA1/2 wildtype prostate cancer cells with MMS22L deletion are hypersensitive to PARPi's via p53-dependent mechanisms. Authors have satisfactorily addressed most of my earlier comments, and there are only some minor issues that remain.

1. In response to Comment 1: It was mentioned that "most missense mutations identified in our cell lines.....protein function". Where are the data or reference(s) that support this statement?
2. In response to Comment 2; authors mentioned that "on the other hand.....genetic background" and argued that LNCaP and C4-2B cells are genetically different cell types. As authors mentioned in their response to Comment 1, the interpretation of genetic variants is not clear-cut in these cells. Were the major differences shown in the two referenced papers here related to DDR and HRD genes?
3. In line 187, it was stated that 'RAD51 loading was enhanced after MMS22L expression was restored'. Where are the data supporting that statement?
4. Line 304: '..... sensitized CHEK-KO C4-2B cells' should be 'CHEK2'.
5. For tumor data in Figure 8c, the endpoint tumor weight should be presented.

Reviewer #2:

Remarks to the Author:

In this revision, the authors carried out additional experiments to investigate the mechanism by which CHEK2 KO confers PARPi resistance in prostate cancer cells. The authors presented evidence that CHEK2 kinase inhibitor does not confer the level of resistance to PARPi as seen in CHEK2 KO cells. This figure should be included in the manuscript, at least as a supplemental figure. This suggests that PARPi resistance in CHEK2 KO cells is likely due to an adaptive response to long-term, rather than acute loss, of CHEK2. If so, the authors should test whether CHEK2 inhibitor leads to decrease in E2F7 and increase in BRCA2 protein levels as seen in the CHEK2 KO cells. It is possible that changes in E2F7 and BRCA2 levels is an indirect, adaptive response to CHEK2 loss, rather than the acute and direct mechanism through p53 phosphorylation as the authors presented. The modest increase in BRCA2 protein level and CHK1 phosphorylation in CHEK2 KO cells remains somewhat unconvincing to explain PARPi resistance.

Reviewer #3:

Remarks to the Author:

I would like to thank for authors for the efforts to address my comments and those from my colleagues. There are still a few issues I would like to raise for the editor consideration:

- I am not confident on their claim for "haploinsufficiency" for MMS22L, which the authors based on their CRISPR experiment, and they use as a rationale to claim the potential clinical impact in a larger population of patients with MMS22L heterozygous loss. To me, this would require thorough validation using other models. The genomic signatures/scores for the subset of patients with MMS22L het-loss does not seem the same as for hom-del either in their figures. I would suggest removing that claim.
- I asked the authors to show the data in Figure 2B in the other models beyond C4-2. Their response that this has led to other findings but they prefer to publish them separately is quite surprising. I still think that showing consistency across cell lines is key.
- The data figure 5, supporting TP53 needs to be intact for MMS22L loss to induce PARPi sensitivity is very interesting. However, if this is true, the authors should adjust their discussion and figures, to show the prevalence of combined MMS22L hom-del/TP53 WT genotype in the cohorts interrogated. How does concomitant loss of TP53 affect genomic signatures?
- The synergistic effect for PARPi-ATRI in preclinical models of prostate cancer has been already

reported, including a paper from the same group. How is the data here different/novel? It does not seem specific to CHEK2 loss models.

- The potential impact of the whole section on CHEK2 loss as resistant biomarker is rather modest; as previously stated, it is difficult to interpret the value of a "induction of resistance" experiment when the model used is not sensitive to the drug in the first place. The authors have tempered the wording in the discussion, which was necessary.

- I would like to add that I stand with my fellow reviewer with regards to the fact that the authors have picked MMSL2 while the data for RNASH2, HELLS and WDR76 in their CRISPR screen is at least equally promising. I understand the degree of effort needed to validate each of them is significant, so I would defer to the editorial position on publishing data for just one of them, but at least, in my opinion, this requires a more clear explanation in the paper. Their previous response is that they picked MMSL2 due to the high prevalence of heterozygous loss, but as I said before, I am not convinced by their claim of haploinsufficiency.

- Most important of all, if the paper is to be accepted, I would really recommend the tone of the discussion is further adjusted, toning down the claims of clinical actionability of MMS22L loss until more data emerges in this space; at most I would say this study raises questions that now need to be tested in clinical trials.

We thank the reviewers again for reviewing our manuscript and helping improve the manuscript. Below we provide a point-by-point response to the reviewers' concerns. Reviewer comments are shown in **Bold**. We have highlighted the changes within the manuscript in red.

Reviewer #1

Authors have satisfactorily addressed most of my earlier comments, and there are only some minor issues that remain.

1. In response to Comment 1: It was mentioned that “most missense mutations identified in our cell lines.....protein function”. Where are the data or reference(s) that support this statement?

It is not trivial to determine the impact of a missense mutation on protein function experimentally. The biological consequences of missense mutations are largely predicted by computational tools, such as Ensemble Variant Effect Predictor (VEP). Accordingly, our statement is not based on any references or experimental validation. Instead, we have used VEP to define deleterious mutation in our study (in Supplementary Data 1). We state that these four cell lines “have no predicted biallelic deleterious mutations” in the manuscript.

2. In response to Comment 2; authors mentioned that “on the other hand.....genetic background” and argued that LNCaP and C4-2B cells are genetically different cell types. As authors mentioned in their response to Comment 1, the interpretation of genetic variants is not clear-cut in these cells. Were the major differences shown in the two referenced papers here related to DDR and HRD genes?

We agree that C4-2B cells are genetically closer to LNCaP cells in contrast to 22Rv1 and DU145 cells that used in our CRISPR screens. The common hits identified between LNCaP and C4-2B cells are likely less valuable than those identified with 22Rv1 and DU145 cells. However, our overall conclusion remains unchanged regardless of how we define common hits. Importantly, in this study, we try to address the question why some of the hits are unique for one cell line, but not for the others. Our results provide evidence suggesting that concurrent genomic alterations may have an antagonistic effect.

In addition, the major difference shown in LNCaP vs. C4-2B cells are not in DDR or HRD genes. However, we don't know whether non-DDR genetic alterations may affect homologous recombination repair function indirectly through unknown mechanisms, which needs further investigation.

3. In line 187, it was stated that ‘RAD51 loading was enhanced after MMS22L expression was restored’. Where are the data supporting that statement?

In Figure 4d (see below), we have shown that RAD51 foci significantly increased after Dox-induced MMS22L expression. In the right panel (blue dots), we observed that the average RAD51 foci increased from 11% to 35% ($p < 0.0001$).

Fig. 4d

Fig. 4d, Representative images and quantification of RAD51 foci in AAVS1 control and MMS22L-KO C4-2B cells stably infected with TET-inducible MMS22L gene after olaparib (5 μ M) treatment in the presence or absence of doxycycline (0.15 μ g/ml) for 24 h. Dots indicate each replicate with more than 100 cells analyzed. Data are presented as mean \pm SD of three biologically independent replicates. Scale bar = 5 μ m.

4. Line 304: ‘..... sensitized CHEK-KO C4-2B cells’ should be ‘CHEK2’.

We thank the reviewer for correcting our mistake. This has been corrected in the text.

5. For tumor data in Figure 8c, the endpoint tumor weight should be presented.

Unfortunately, we didn’t used tumor weight as an endpoint in this study. The drug response is largely based on the tumor volume.

Reviewer #2

In this revision, the authors carried out additional experiments to investigate the mechanism by which CHEK2 KO confers PARPi resistance in prostate cancer cells. The authors presented evidence that CHEK2 kinase inhibitor does not confer the level of resistance to PARPi as seen in CHEK2 KO cells. This figure should be included in the manuscript, at least as a supplemental figure. This suggests that PARPi resistance in CHEK2 KO cells is likely due to an adaptive response to long-term, rather than acute loss, of CHEK2. If so, the authors should test whether CHEK2 inhibitor leads to decrease in E2F7 and increase in BRCA2 protein levels as seen in the CHEK2 KO cells. It is possible that changes in E2F7 and BRCA2 levels is an indirect, adaptive response to CHEK2 loss, rather than the acute and direct mechanism through p53 phosphorylation as the authors presented. The modest increase in BRCA2 protein level and CHK1 phosphorylation in CHEK2 KO cells remains somewhat unconvincing to explain PARPi resistance.

We have provided evidence including ChIP and gene expression data, supporting that BRCA2 is a target of CHEK2-TP53-E2F7-mediated transcriptional repression. This at least partially explains the mechanism of enhanced HRR capacity after CHEK2 loss. However, we agree that the mechanism of CHEK2 loss mediated PARPi resistance is incomplete. As the reviewer pointed out, it is not clear whether the resistance in CHEK2 KO cells is due to an adaptive response to long-term or acute loss. We have examined E2F7 and BRCA2 expression after CHEK2 inhibitor BML277 treatment. The changes in gene expression are subtle in line with the result, showing that BML277 has little impact on olaparib response. However, we think that further investigation is needed to fully understand the mechanism.

We need to carefully design the CHEK2 inhibitor experiments with different does, time, and cells. We also need to determine whether BML277 inhibits BRCA1 phosphorylation and protein stability and whether BML277 has off-target effects. More importantly, we need to determine

whether CHEK2 inhibitor affects homologous recombination repair function, such as RAD51 foci formation. In addition, siRNA knockdown is another approach to determine the effects of acute loss. This requires more time and effort. That said, the significance of our finding is that CHEK2 mutations may not be used as a biomarker for PARP inhibition, supporting a reevaluation of current biomarkers for the use of PARP inhibitors in mCRPC patients.

Reviewer #3

I would like to thank for authors for the efforts to address my comments and those from my colleagues. There are still a few issues I would like to raise for the editor consideration:

- I am not confident on their claim for “haploinsufficiency” for MMS22L, which the authors based on their CRISPR experiment, and they use as a rationale to claim the potential clinical impact in a larger population of patients with MMS22L heterozygous loss. To me, this would require throughout validation using other models. The genomic signatures/scores for the subset of patients with MMS22L het-loss does not seem the same as for hom-del either in their figures. I would suggest removing that claim.

We agree. We have removed the claim for “haploinsufficiency” of MMS22L. We only state that “all MMS22L-KO clones exhibited high sensitivity to olaparib to a similar extent without a gene dose effect.”

- I asked the authors to show the data in Figure 2B in the other models beyond C4-2. Their response that this has led to other findings, but they prefer to publish them separately is quite surprising. I still think that showing consistency across cell lines is key.

It should be noted that the genes validated in C4-2B cells in Figure 2B are not the common hits in all four cell lines. We now have provided extra validation data for WDR76 and HELLS genes that are common hits in all four cell lines (see Supplementary Fig 5b).

Figure S5b. Dose-response curves after treatment with olaparib for AAVS1 control, HELLS-KO, and WDR76-KO 22RV1 and DU145 cell clones. Immunoblot analysis of the HELLS and WDR76 protein levels in AAVS1 control, HELLS-KO, and WDR76-KO cell clones are shown. Two KO cell clones (cl) were selected for each cell viability experiment as indicated. Data are presented as mean \pm SD of six biologically independent replicates. The *p*-values were determined by comparing two gene-specific KO cell clones to a control AAVS1 cell clone using two-way ANOVA.

- The data figure 5, supporting TP53 needs to be intact for MMS22L loss to induce PARPi sensitivity is very interesting. However, if this is true, the authors should adjust their discussion and figures, to show the prevalence of combined MM22L hom-del/TP53 WT genotype in the cohorts interrogated. How does concomitant loss of TP53 affects genomic signatures?

As the reviewer suggested, we now have shown the prevalence of MM22L homo-deletion/TP53 WT genotype in Figure 5i. We added the following in the text:

“Approximately 12.6% of primary and 3.8% of metastatic prostate tumors harbor MMS22L homozygous deletion with wild-type TP53 (Fig. 5i). The percentage will increase to 27% and 20.5% respectively when heterozygous deletion is included.”

In addition, loss of TP53 alone has little effect on genomic signature or HRD score. However, the number of cases in the cohort is too small to determine whether and to what extent concomitant loss of TP53 and MMS22L affects genomic signature.

- The synergistic effect for PARPi-ATRi in preclinical models of prostate cancer has been already reported, including a paper from the same group. How is the data here different/novel? It does not seem specific to CHEK2 loss models.

We agree that the PARPi-ATRi combination therapy is not novel or specific to CHEK2 loss models. Here, we intend to demonstrate that PARPi resistance caused by CHEK2 loss can be treated with the combination as well. We have revised this in the discussion:

“Indeed, the combination of PARPi with ATRi was synergistic in many PARPi-resistant models regardless of their resistance mechanisms. Our previous studies support the use of combined PARP and ATR inhibition for PARPi-resistant prostate tumors with RB1 loss. Simultaneous suppression of ATR and PARP may be a preferred strategy to overcome PARPi resistance.”

- The potential impact of the whole section on CHEK2 loss as resistant biomarker is rather modest; as previously stated, it is difficult to interpret the value of a “induction of resistance” experiment when the model used is not sensitive to the drug in the first place. The authors have tempered the wording in the discussion, which was necessary.

We agree that the mechanism of CHEK2 loss-mediated resistance is still incomplete. As we have addressed the reviewer #2’s concern, further investigation is needed. However, our finding supports a reevaluation of CHEK2 as a biomarker for the use of PARP inhibition in mCRPC patients.

- I would like to add that I stand with my fellow reviewer with regards to the fact that the authors have picked MMSL2 while the data for RNASEH2, HELLS and WDR76 in their CRISPR screen is at least equally promising. I understand the degree of effort needed to validate each of them is significant, so I would defer to the editorial position on publishing data for just one of them, but at least, in my opinion, this requires a more clear explanation in the paper. Their previous response is that they picked MMSL2 due to the high prevalence of heterozygous loss, but as I said before, I am not convinced by their claim of haploinsufficiency.

RNASEH2B has been well-studied in previous work including our own (Zimmermann et al, Nature 2018, PMID 29973717 and Miao et al, Sci Adv 2022, PMID 35179959). We now have added new validation data for the HELLS and WDR76 genes although the mechanistic studies are still ongoing (new Supplementary Fig. 5b).

- Most important of all, if the paper is to be accepted, I would really recommend the tone of the discussion is further adjusted, toning down the claims of clinical actionability of MMS22L loss until more data emerges in this space; at most I would say this study raises questions that now need to be tested in clinical trials.

We agree that our findings heavily rely on pre-clinical models. We now have toned down the claim in the discussion. We added that “these results from preclinical models are still subject to clinical validation....”